# Meta-iLaD: Identifiable Latent Dynamics via Meta-Learning of Dynamics Environments

Yubo Ye [1]  Sweekar Piya [2]  Xiajun Jiang [2]  Linwei Wang [1]

## Abstract

Learning *latent dynamics* is central to assessing the current states and forecasting the future trajectories of high-dimensional time series. For locally-stationary latent dynamics $\mathcal{F}(\mathbf{z}_{<t}; \mathbf{c})$ with latent dynamics state $\mathbf{z}_t$ and environment variable $\mathbf{c}$, prior identifiability results have largely focused on $\mathbf{z}_t$ when conditioned on predefined label $u$ of dynamics environments. This leaves two limitations: reliance on predefined labels that hinder generalization to unseen environments, and a limited understanding of the identifiability of $\mathcal{F}$ and $\mathbf{c}$ which—although offering important structural properties for the identifiability of $\mathbf{z}_t$—are learned jointly with $\mathbf{z}_t$. We address these limitations with Meta-iLaD, a novel identifiable latent dynamics framework attained by meta-learning across dynamics environments. Meta-iLaD introduces a novel conditional prior of $\mathbf{c}$, modeled as a feedforward meta-learner to rapidly extract $\mathbf{c}$ from few-shot examples. Meta-iLaD further establishes identifiability for $\mathbf{z}_t$, $\mathbf{c}$, and $\mathcal{F}$ simultaneously, for a general formulation of $\mathcal{F}(\mathbf{z}_{<t}; \mathbf{c})$ without restricting the dimension of $\mathbf{c}$ or how it modulates $\mathcal{F}$. On synthetic and real data, we provide strong empirical evidence that 1) conditioning on few-shot examples enables generalization to out-of-distribution environments, and 2) identifiability for $\mathbf{c}$ and $\mathcal{F}$ is critical for accurate forecasting beyond reconstructing observed trajectories.

[1]Department of Computing and Information Sciences, Rochester Institute of Technology, Rochester, New York, USA [2]Department of Computer Science, Rowan University, Glassboro, New Jersey, USA. Correspondence to: Yubo Ye <yy8339@rit.edu>.

*Proceedings of the 43rd International Conference on Machine Learning*, Seoul, South Korea. PMLR 306, 2026. Copyright 2026 by the author(s).

## 1. Introduction

Many domains such as healthcare are observing increasingly high-dimensional (*e.g.,* image or multi-modal) data over time. It is important to extract the latent dynamics that governs the observed data, not only for assessing the current states of the system, but also for forecasting its future trajectories. An important approach is to model the dynamics in a low-dimensional latent space as illustrated in Fig. 1(a): the observed $\mathbf{x}_t$'s is generated from the *latent dynamics state* $\mathbf{z}_t$ via an *emission function* $\mathbf{x}_t = \mathbf{g}(\mathbf{z}_t)$; the temporal transition of $\mathbf{z}_t$ is governed by a *latent dynamics function* $\mathbf{z}_t = \mathcal{F}(\mathbf{z}_{<t}; \mathbf{c})$, where a *dynamics environment variable* $\mathbf{c}$ allows non-stationary $\mathcal{F}$ (but can be omitted otherwise). The goal of *latent dynamics modeling* is to learn $\mathcal{F}$ and $\mathbf{g}$ from observed $\mathbf{x}_t$'s *without* direct supervisions on $\mathbf{z}_t$'s.

While significant progress has been made in this research direction (Krishnan et al., 2017; Fraccaro et al., 2017; Yıldız et al., 2019; Botev et al., 2021), we are interested in a specific question: even when $p(\hat{\mathbf{x}}_t)$ is perfectly optimized to be equivalent to the observed $p(\mathbf{x}_t)$, what can we say about its underlying $\hat{\mathbf{z}}_t$, $\hat{\mathbf{c}}$, and $\hat{\mathcal{F}}$ in relation to their respective truth? This question of *identifiability*, as outlined in Fig. 1(a), exists at three levels: identifiability of the latent dynamics state $\mathbf{z}_t$, identifiability of the latent dynamics function $\mathcal{F}$, and identifiability of the dynamics environment variable $\mathbf{c}$.

**Identifiability of latent dynamics states:** Identifiability of the latent state $\mathbf{z}_t$ has been the primary focus of existing works in identifiable latent dynamics. Successes are mainly built on the theory of nonlinear independent component analysis (ICA) (Hyvärinen et al., 2019; Khemakhem et al., 2020a), across three types of latent dynamics systems: stationary $\mathcal{F}(\mathbf{z}_{<t})$ (Hyvärinen & Morioka, 2017; Klindt et al., 2021), non-stationary $\mathcal{F}(\mathbf{z}_{<t}; \mathbf{c}_t)$ where $\mathbf{c}_t$ follows a hidden Markov model (Song et al., 2023; Li et al., 2024; Hälvä & Hyvärinen, 2020; Hälvä et al., 2021; Balsells-Rodas et al., 2024), and locally-stationary $\mathcal{F}(\mathbf{z}_{<t}; \mathbf{c})$ where $\mathbf{c}$ characterizes the underlying dynamics environments (Khemakhem et al., 2020a; Yao et al., 2021; 2022; Hızlı et al., 2025). Our primary interest is in the last category of locally-stationary dynamics $\mathcal{F}(\mathbf{z}_{<t}; \mathbf{c})$, where two critical gaps remain.

First, existing works in $\mathcal{F}(\mathbf{z}_{<t}; \mathbf{c})$ establish the identifiabil-

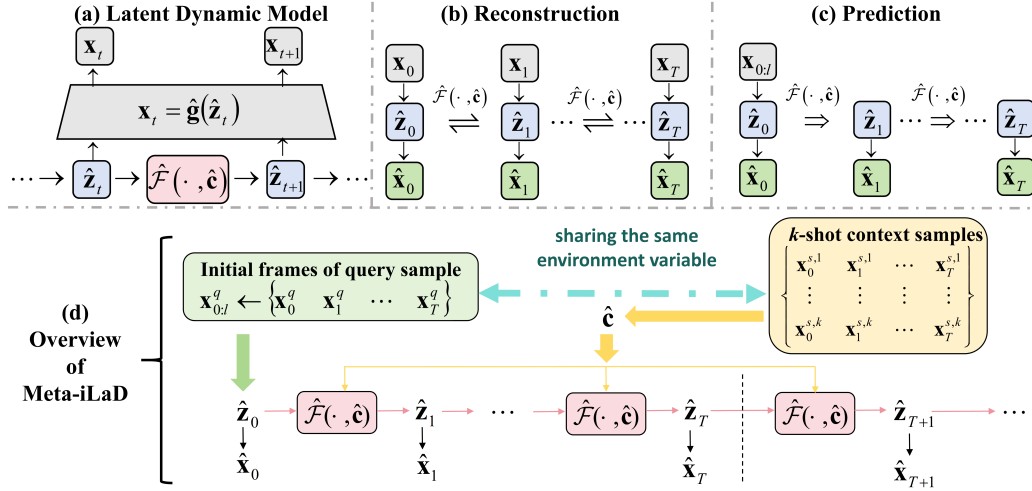

*Figure 1.* Overview of a latent dynamics system with latent dynamics state $\mathbf{z}_t$, environment variable $\mathbf{c}$, and dynamics function $\mathcal{F}$ (a), its use in reconstruction (b) *vs.* prediction (c) settings, and Meta-iLaD which establishes $\mathbf{z}_t = \mathcal{F}(\mathbf{z}_{<t}; \mathbf{c})$ with identifiable $\mathbf{z}_t$, $\mathbf{c}$ and $\mathcal{F}$ (d).

ity of $\mathbf{z}_t$'s by conditioning $\mathcal{F}$ on observed labels $u$ of the dynamics environments. Unless true parameters characterizing such environments are available, $u$ needs to be predefined as discrete environment labels. This raises fundamental questions regarding 1) generalization to out-of-distribution (OOD) environments not included in the training labels, and 2) applicability when such label $u$ is absent.

Second, while $\mathcal{F}(\mathbf{z}_{<t}; \mathbf{c})$ is leveraged as important structural properties to establish the identifiability of $\mathbf{z}_t$, the identifiability of $\mathcal{F}$ and $\mathbf{c}$ themselves has only been examined in one recent work (Hızlı et al., 2025) even though they are learned simultaneously with $\mathbf{z}_t$. This is largely inconsequential in the *reconstruction* setting considered in most existing works where, as illustrated in Fig. 1(b), one observes a sequence of $\mathbf{x}_{0:T}$'s and extract latent $\hat{\mathbf{z}}_t$'s to reconstruct $\hat{\mathbf{x}}_{0:T}$. There is however a distinct yet equally important *prediction* setting where, as illustrated in Fig. 1(c), one observes $\mathbf{x}_{0:l}$ and uses learned latent dynamics function $\hat{\mathcal{F}}(\hat{\mathbf{z}}_{<t}; \hat{\mathbf{c}})$ to predict future $\hat{\mathbf{z}}_t$ and $\hat{\mathbf{x}}_t$ for $t > l$. Because future latent states are evolved directly through $\hat{\mathcal{F}}(\hat{\mathbf{z}}_{<t}; \hat{\mathbf{c}})$ in this setting, the identifiability of $\mathcal{F}$ and $\mathbf{c}$ becomes more important yet less studied.

**Identifiability of latent dynamics environments & functions:** In the general literature of latent dynamics modeling, increasing works have taken interest in inferring the latent environment variable $\mathbf{c}$ underlying locally-stationary $\mathcal{F}(\mathbf{z}_{<t}; \mathbf{c})$ (Auzina et al., 2023; Wang et al., 2022; Kirchmeyer et al., 2022; Jiang et al., 2022). These works however do not investigate or establish the identifiability for $\mathbf{c}$, nor its relation to the identifiability of dynamics function $\mathcal{F}$.

In the literature of identifiable latent dynamics, the identifiability of latent environment variable $\mathbf{c}$ has been mainly studied in the category of non-stationary $\mathcal{F}(\mathbf{z}_{<t}; \mathbf{c}_t)$, assuming $\mathbf{c}_t$ to be the hidden state of an HMM process (Song et al., 2023; Li et al., 2024; Hälvä & Hyvärinen, 2020;

Hälvä et al., 2021; Balsells-Rodas et al., 2024). For locally-stationary $\mathcal{F}(\mathbf{z}_{<t}; \mathbf{c})$, a recent work (Hızlı et al., 2025) took the first step through by introducing a process noise $\mathbf{s}_t$ and assuming it to be factorized when conditioned on predefined environment label $u$: with $\mathcal{F}(\mathbf{z}_{<t}, \mathbf{s}_t|u)$, identifiability is established simultaneously for $\mathbf{z}_t$, $\mathbf{s}_t$, and $\mathcal{F}$. While an important first step, the use of process noise $\mathbf{s}_t$ represents a specific assumption of how latent dynamics is modulated by its environments. It is associated with several restrictive conditions, *e.g.*, $\mathbf{s}_t$ must match the dimensionality of $\mathbf{z}_t$ and have a *dimension-wise* influence on $\mathcal{F}$ (each dimension of $\mathcal{F}$ is influenced by a single dimension noise variable in $\mathbf{s}_t$).

**Contribution:** We introduce Meta-iLaD, a novel framework for identifiable latent dynamics $\mathcal{F}(\mathbf{z}_{<t}; \mathbf{c})$ via meta-learning across dynamics environments. As outlined in Fig. 1(d), instead of conditioning on predefined environment labels, Meta-iLaD conditions $\mathcal{F}(\mathbf{z}_{<t}; \mathbf{c})$ on few-shot context examples from a dynamics environment, from which the latent environment variable $\mathbf{c}$ is inferred. It advances identifiability research with the following contributions:

- **Theoretically**, we establish the first simultaneous identifiability results for latent state $\mathbf{z}_t$, latent environment variable $\mathbf{c}$ (including its dimensionality), and latent dynamics function $\mathcal{F}$ for a general formulation of $\mathcal{F}(\mathbf{z}_{<t}; \mathbf{c})$, where the latent environment variable $\mathbf{c}$ is not limited to a specific dimension or form, but can modulate $\mathcal{F}$ in a general and interpretable fashion, *e.g.*, as parameters characterizing the dynamics environments. Furthermore, this is the first identifiable $\mathcal{F}(\mathbf{z}_{<t}; \mathbf{c})$ that is conditioned on few-shot context samples rather than predefined environment labels.

- **Methodologically**, we introduce a novel formulation of the conditional prior of $\mathbf{c}$, as a feedforward meta-learner that rapidly extracts $\mathbf{c}$ from few-shot context examples from an environment. This conditional latent

environment variable then adapts the latent $\mathcal{F}$ to evolve latent dynamics states $\mathbf{z}_t$'s to emit to $\mathbf{x}_t$'s specific to that environment, establishing meta-learning as a novel solution to attain identifiable latent dynamics.

- **Experimentally**, we provide strong empirical evidence for the two key theoretical contributions of Meta-iLaD. First, compared to predefined environment labels $u$'s, we show that conditioning $\mathbf{c}$ on few-shot examples enables Meta-iLaD to generalize to OOD environments as well as apply to a broader variety of systems where environment labels $u$ are difficult to define. Second, moving beyond reconstruction-focused evaluation settings commonly adopted in prior work, we provide evidence that identifiability of the environment variable $\mathbf{c}$ and dynamics function $\mathcal{F}$ are critical for predicting beyond observed trajectories.

## 2. Related Works

### 2.1. Identifiable Latent Dynamics Models

Recent advances in nonlinear ICA theory have established that identifiability of a latent variable model can be attained by constructing its conditional independence given auxiliary information (Hyvärinen & Morioka, 2016; 2017; Hyvärinen et al., 2019; Khemakhem et al., 2020a). This theoretical foundation has enabled rapid developments in identifiable latent dynamics models, primarily focused on identifiability of $\mathbf{z}_t$ for three types of latent dynamics.

**Identifiability of latent states.** For **stationary dynamics** $\mathcal{F}(\mathbf{z}_{<t})$, identifiability of $\mathbf{z}_t$ has been established by its temporal dependency (Hyvärinen & Morioka, 2017; Klindt et al., 2021). For **non-stationary dynamics** $\mathcal{F}(\mathbf{z}_{<t}; \mathbf{c}_t)$, identifiability of $\mathbf{z}_t$ has been established when $\mathbf{c}_t$ represents the hidden state of a first-order Markov process ($\mathbf{c}_t \sim \text{Markov Chain}(\mathbf{A})$). Different forms of $\mathcal{F}$ have been considered, including no additional temporal transition beyond the HMM (Hälvä & Hyvärinen, 2020; Balsells-Rodas et al., 2024), linear transitions (Hälvä et al., 2021), and non-parametric transitions (Song et al., 2023). For **locally-stationary dynamics** $\mathbf{z}_t = \mathcal{F}(\mathbf{z}_{<t}; \mathbf{c})$ where $\mathbf{c}$ governs a time segment, identifiability of $\mathbf{z}_t$ has been established for linear (Yao et al., 2021), non-parametric (Yao et al., 2021; 2022), and nonlinear forms of $\mathcal{F}$ (Hızlı et al., 2025). Critically, all these approaches conditions $\mathcal{F}$ on a predefined label $u$ of the dynamics environment (Khemakhem et al., 2020a; Yao et al., 2021; 2022; Hızlı et al., 2025).

**Identifiability of dynamics functions.** Most existing works focus solely on identifiability of $\mathbf{z}_t$. A recent exception (Hızlı et al., 2025) establishes identifiability for both $\mathbf{z}_t$ and $\mathcal{F}$ by introducing a conditionally-factorized process noise $\mathbf{s}_t$ with $p(\mathbf{s}_t|u)$. This assumes the environment to modulate each dimension of $\mathcal{F}$ via a scalar noise, requiring $\mathbf{s}_t$ to match

the dimension of $\mathbf{z}_t$ and still relies on predefined labels $u$.

**Position of our work.** We advance locally-stationary latent dynamics $\mathcal{F}(\mathbf{z}_{<t}; \mathbf{c})$ in two key aspects. First, we extend identifiability to the latent environment variable $\mathbf{c}$ and dynamics function $\mathcal{F}$ for a general formulation of $\mathcal{F}(\mathbf{z}_{<t}; \mathbf{c})$. It goes beyond the assumptions of dimension-wise modulation of $\mathcal{F}$ and does not restrict the dimensionality of $\mathbf{c}$. Second, while the general formulation of $\mathcal{F}(\mathbf{z}_{<t}; \mathbf{c})$ exists (Yao et al., 2022), we condition it on few-shot data examples instead of predefined environment labels, and show that this allows generalization to OOD environments as well as systems without available environment labels.

### 2.2. Non-Identifiable Latent Dynamics Models

Outside the identifiability literature, significant progress has been made in learning latent dynamics for forecasting high-dimensional time series (Chung et al., 2015; Krishnan et al., 2017; Karl et al., 2016; Fraccaro et al., 2017; Becker-Ehmck et al., 2019; Linderman et al., 2017; Rubanova et al., 2019; Yıldız et al., 2019; Botev et al., 2021). These methods prioritize predictive performance but do not establish identifiability for the learned latent states or dynamics functions.

**Environment-adaptive dynamics.** There has been a rising interest in inferring latent environment variables to adapt latent dynamics functions (Auzina et al., 2023; Wang et al., 2022; Jiang et al., 2022; Kirchmeyer et al., 2022). While these approaches demonstrate improved forecasting by capturing environment-specific dynamics, they do not discuss the identifiability of the latent environment variable or its relationship to the recoverability of the underlying dynamics.

**Position of our work.** In contrast, Meta-iLaD establishes identifiability for both the latent environment variable $\mathbf{c}$ and the dynamics function $\mathcal{F}$, introduces meta-learning as a principled solution to attain this identifiability, and provides empirical evidence that such identifiability is critical for accurate forecasting performance beyond reconstruction.

## 3. Identifiable Latent Dynamics via Meta-iLaD

Consider $n$-dimensional time-series data $\mathbf{x}_t \in \mathbb{R}^n$ at time $t$, generated from $m$-dimensional latent variable $\mathbf{z}_t \in \mathbb{R}^m$ via an invertible mixing function $\mathbf{g}$. We assume that the dynamics of $\mathbf{z}_t$ is governed by the transition function $\mathcal{F}$ and the environment variable $\mathbf{c} \in \mathbb{R}^d$:

$$\mathbf{z}_t = \mathcal{F}(\mathbf{z}_{t-1}; \mathbf{c}); \quad \mathbf{x}_t = \mathbf{g}(\mathbf{z}_t) + \epsilon, \quad 0 \le t \le T \quad (1)$$

where $\epsilon \sim p_\epsilon(\epsilon)$ is an independent noise. While $\mathcal{F}$ can be described by a variety of functions, we consider a neural ordinary differential equation (NODE) $\mathbf{f}_\theta$:

$$\frac{dz_{i,t}}{dt} = f_{i,\theta}(\mathbf{z}_t; \mathbf{c}) \Leftrightarrow \frac{d\mathbf{z}_t}{dt} = \mathbf{f}_\theta(\mathbf{z}_t; \mathbf{c}), \quad (2)$$

where $z_{i,t}$ is the $i$-th component in $\mathbf{z}_t$. $\mathbf{f}_\theta$ is parameterized by $\theta$ and locally stationary dependent on $\mathbf{c}$. An unconditional generative model for this system is:

$$
\begin{aligned}
p_\Theta(\mathbf{x}_{0:T}) = \int_\mathbf{c} \int_{\mathbf{z}_{0:T}} &p(\mathbf{c})p(\mathbf{z}_0) \prod_{t=1}^{T} p_\theta(\mathbf{z}_t|\mathbf{z}_{t-1}, \mathbf{c}) \\
&\times \prod_{t=0}^{T} p_\mathbf{g}(\mathbf{x}_t|\mathbf{z}_t)d\mathbf{z}_{0:T}d\mathbf{c},
\end{aligned} \quad (3)
$$

where $\Theta = \{\theta, \mathbf{g}\}$ and $p_\mathbf{g}(\mathbf{x}_t|\mathbf{z}_t) = p_\epsilon(\mathbf{x}_t - \mathbf{g}(\mathbf{z}_t))$.

Unfortunately, with this unconditional generative model, neither the environment variable $\mathbf{c}$ nor the latent variables $\mathbf{z}_t$ can be uniquely recovered from observations $\mathbf{x}_{0:T}$ alone. To address this identifiability issue, we construct a conditional generative model $p_\Phi(\mathbf{x}_{0:T}|\mathbf{u})$, where $\mathbf{u}$ consists of $k$-shot samples that share the same environment variable $\mathbf{c}$ as $\mathbf{x}_{0:T}$ but are otherwise distinct. For clarity, in Section 3.1, we define $\mathbf{u} := \mathcal{D}^s$ and refer to the samples in $\mathcal{D}^s$ as *context* samples (for inferring $\mathbf{c}$); we also define $\mathbf{x}_{0:T} := \mathbf{x}_{0:T}^q$ and refer to it as the *query sample* (to be generated). The conditional density $p_\Phi(\mathbf{x}_{0:T}|\mathbf{u}) := p_\Phi(\mathbf{x}_{0:T}^q|\mathcal{D}^s)$ thus defines the marginal likelihood of a query sample $\mathbf{x}_{0:T}^q$ conditioned on distinct context samples from the same environment.

### 3.1. Construction of Identifiable Latent Dynamics

**Latent environment variable c:** Given context samples $\mathcal{D}^s$ from the same dynamics environment, we define the conditional density for its latent environment variable $\mathbf{c}$ as:

$$
\begin{aligned}
p_\zeta(\mathbf{c}|\mathcal{D}^s) &= \prod_{i=1}^{d} \mathcal{N}(c_i|\mu_{i,\zeta}(\mathcal{D}^s), \sigma_{i,\zeta}(\mathcal{D}^s)) \\
&= \mathcal{N}(\mathbf{c}|\boldsymbol{\mu}_\zeta(\mathcal{D}^s), \boldsymbol{\sigma}_\zeta(\mathcal{D}^s)),
\end{aligned} \quad (4)
$$

where the $d$-dimensional $\mathbf{c}$ is assumed to be conditionally factorized and each element $c_i \in \mathbf{c}$ is Gaussian. The multivariate mean $\boldsymbol{\mu}_\zeta(\mathcal{D}^s)$ and variance $\boldsymbol{\sigma}_\zeta(\mathcal{D}^s)$ are realized as:

$$
\begin{aligned}
\boldsymbol{\mu}_\zeta(\mathcal{D}^s) &= \frac{1}{|\mathcal{D}^s|} \sum_{\mathbf{x}_{0:T}^s \in \mathcal{D}^s} \boldsymbol{h}_{\mu,\zeta}(\mathbf{x}_{0:T}^s), \\
\boldsymbol{\sigma}_\zeta(\mathcal{D}^s) &= \frac{1}{|\mathcal{D}^s|} \sum_{\mathbf{x}_{0:T}^s \in \mathcal{D}^s} \boldsymbol{h}_{\sigma,\zeta}(\mathbf{x}_{0:T}^s),
\end{aligned} \quad (5)
$$

where $\boldsymbol{h}_{\mu,\zeta}$ and $\boldsymbol{h}_{\sigma,\zeta}$ represent the two heads of an encoding network parameterized by $\zeta$, and the averaging function extracts the embedding shared by the $k$-shot context samples.

**Latent dynamics state $\mathbf{z}_t$:** When conditioned on $\mathbf{c}$, we define the transition probability of $\mathbf{z}_t$ as:

$$
p_\theta(\mathbf{z}_t|\mathbf{z}_{t-1}, \mathbf{c}) = \delta(\mathbf{z}_t - \mathcal{F}_\theta(\mathbf{z}_{t-1}, \mathbf{c})),
$$

$$
\mathcal{F}_\theta(\mathbf{z}_{t-1}, \mathbf{c}) = \mathbf{z}_{t-1} + \int_{s=t-1}^{t} \mathbf{f}_\theta(\mathbf{z}_s; \mathbf{c})ds.
$$

In other words, we assume the stochasticity in $\mathcal{F}$ to be explained by the environment variable $\mathbf{c}$ and $\mathbf{z}_0$. Furthermore,

when conditioned on $\mathbf{c}$ and $\mathbf{z}_{t-1}$, $\mathbf{z}_t$ is assumed factorized as $p_\theta(\mathbf{z}_t|\mathbf{z}_{t-1}, \mathbf{c}) = \prod_{i=1}^{m} p_\theta(z_{i,t}|\mathbf{z}_{t-1}, \mathbf{c})$.

To marginalize out $\mathbf{z}_{t-1}$ from $p_\theta(\mathbf{z}_t|\mathbf{z}_{t-1}, \mathbf{c})$, $p_\theta(\mathbf{z}_t|\mathbf{c})$ can also be derived from $p(\mathbf{z}_0)$ through the NODE flow in Equation (2) via the instantaneous change of variables:

$$
\log p_\theta(\mathbf{z}_t|\mathbf{c}) = \log p(\mathbf{z}_0) - \int_0^t \text{tr}\left(J_{\mathbf{f}_\theta}(\mathbf{z}_\tau, \mathbf{c})\right) d\tau, \quad (6)
$$

where $J_{\mathbf{f}_\theta}(\mathbf{z}_t, \mathbf{c}) = \frac{\partial \mathbf{f}_\theta(\mathbf{z}_t, \mathbf{c})}{\partial \mathbf{z}_t}$ denotes the Jacobian matrix.

**Conditional generative models:** For a query sample $\mathbf{x}_{0:T}^q$ from the same environment as the context samples $\mathcal{D}^s$, we now extend the unconditonal generative model in Equation (3) to a conditional $p_\Phi(\mathbf{x}_{0:T}^q|\mathcal{D}^s)$ with $\Phi = (\Theta, \zeta)$ as:

$$
\begin{aligned}
p_\Phi(\mathbf{x}_{0:T}^q|\mathcal{D}^s) &= \int_\mathbf{c} \int_{\mathbf{z}_0} p_\Theta(\mathbf{x}_{0:T}^q|\mathbf{c}, \mathbf{z}_0)p_\zeta(\mathbf{c}|\mathcal{D}^s)p(\mathbf{z}_0)d\mathbf{c}d\mathbf{z}_0 \\
&= \int_\mathbf{c} \int_{\mathbf{z}_0} p_g(\mathbf{x}_0|\mathbf{z}_0) \prod_{t=1}^{T} p_g(\mathbf{x}_t|\mathbf{z}_t)|_{\mathbf{z}_t = \mathcal{F}_\theta(\mathbf{z}_{t-1};\mathbf{c})} \\
&\quad \times p_\zeta(\mathbf{c}|\mathcal{D}^s)p(\mathbf{z}_0)d\mathbf{c}d\mathbf{z}_0.
\end{aligned}
$$

### 3.2. Identifiability Theory

For simplicity in notation, in this section, we again denote $\mathbf{x}_{0:T}^q$ as $\mathbf{x}_{0:T}$ and $\mathcal{D}^s$ as $\mathbf{u}$. We define:

$$
\begin{aligned}
\mathbf{x}_t &= \mathbf{g}\left(\mathbf{z}_0 + \int_{s=0}^{t} \mathbf{f}_\theta(\mathbf{z}_s, \mathbf{c})ds\right) + \epsilon_t \\
&= F_{\mathbf{g},\theta,t}(\mathbf{z}_0, \mathbf{c}) + \epsilon_t,
\end{aligned} \quad (7)
$$

which gives $\mathbf{x}_{0:T} = \mathbf{F}_{\mathbf{g},\theta}(\mathbf{z}_0, \mathbf{c}) + \boldsymbol{\epsilon}$, where $\mathbf{F}_{\mathbf{g},\theta}(\mathbf{z}_0, \mathbf{c}) = [F_{\mathbf{g},\theta,0}(\mathbf{z}_0, \mathbf{c}), \ldots, F_{\mathbf{g},\theta,T}(\mathbf{z}_0, \mathbf{c})]^T$ and $\boldsymbol{\epsilon} = [\epsilon_0, \ldots, \epsilon_T]^T$.

For the prior $p_\zeta(\mathbf{c}|\mathbf{u})$, we define sufficient statistics $\mathbf{T}(\mathbf{c}) = (\mathbf{T}_1(c_1), \ldots, \mathbf{T}_d(c_d))$ with $\mathbf{T}_i(c_i) = (c_i, c_i^2)$, and natural parameters $\boldsymbol{\lambda}_\zeta(\mathbf{u}) = (\boldsymbol{\lambda}_{\zeta 1}(\mathbf{u}), \ldots, \boldsymbol{\lambda}_{\zeta d}(\mathbf{u}))$ where

$$
\boldsymbol{\lambda}_{\zeta i}(\mathbf{u}) = \left(\boldsymbol{\mu}_{\zeta_i}(\mathbf{u})/\boldsymbol{\sigma}_{\zeta_i}^2(\mathbf{u}), -1/2\boldsymbol{\sigma}_{\zeta_i}^2(\mathbf{u})\right). \quad (8)
$$

#### 3.2.1. IDENTIFIABILITY OF ENVIRONMENT VARIABLE c

**Definition 1:** *Let $\mathbf{x}_{0:T}$ be an observed time series generated by Equations (4-7) with parameter $\Phi = (\mathbf{g}, \theta, \zeta)$ and latent environment variable $\mathbf{c}$. A model with parameter $\hat{\Phi} = (\hat{\mathbf{g}}, \hat{\theta}, \hat{\zeta})$ and latent variable $\hat{\mathbf{c}}$ is observationally equivalent to $\Phi$ if $p_{\hat{\Phi}}(\mathbf{x}_{0:T})$ matches $p_\Phi(\mathbf{x}_{0:T})$ everywhere. Let $\sim$ be an equivalence relation on $\Phi$ (see Definition 2), we say that (7) is $\sim$ identifiable if*

$$
p_\Phi(\mathbf{x}_{0:T}|\mathbf{u}) = p_{\hat{\Phi}}(\mathbf{x}_{0:T}|\mathbf{u}) \Rightarrow \Phi \sim \hat{\Phi}. \quad (9)
$$

**Definition 2**: *Let $\sim$ be the equivalence relation on $\Phi$ as:*

$$
(\mathbf{g}, \theta, \zeta) \sim (\hat{\mathbf{g}}, \hat{\theta}, \hat{\zeta}) \Leftrightarrow \exists A, \mathbf{b}, P \mid \mathbf{c} = A \cdot P(\hat{\mathbf{c}}) + \mathbf{b}, \quad (10)
$$

where $A = diag(a_1, \ldots, a_d)$ with $a_i \neq 0$ is a diagonal scaling matrix, $\mathbf{b} \in \mathbb{R}^d$ is a translation vector, and $P$ is a permutation matrix. This means that the true latent environment variable $\mathbf{c}$ is recovered up to permutation, scaling, and translation as the variable in its estimated latent space.

**Theorem 1:** *Assume that we observe data generated according to Equations (4-7) with parameter $\Phi = (\mathbf{g}, \theta, \zeta)$ and latent variable $\mathbf{c}$. Assume the following holds:*

1. *The set $\{\mathbf{x}_{0:T} \in \mathcal{X}^{T+1} | \varphi_\epsilon(\mathbf{x}_{0:T}) = 0\}$ has measure zero, where $\varphi_\epsilon$ is the characteristic function of the density $p_\epsilon$.*

2. *The mixing function $\mathbf{F}_{\mathbf{g},\theta}$ is injective.*

3. *The sufficient statistics $T_{ij}$ are differentiable almost everywhere, and $(T_{ij})_{1 \leq j \leq 2}$ are linearly independent on any subset of $\mathcal{X}^{T+1}$ of measure greater than zero.*

4. *There exist at least $2d + 1$ distinct $\mathbf{u}_0, \ldots, \mathbf{u}_{2d}$ such that the $2d \times 2d$ matrix below is invertible*

$$\mathbf{L} = (\boldsymbol{\lambda}_\zeta(\mathbf{u}_1) - \boldsymbol{\lambda}_\zeta(\mathbf{u}_0), \ldots, \boldsymbol{\lambda}_\zeta(\mathbf{u}_{2d}) - \boldsymbol{\lambda}_\zeta(\mathbf{u}_0)), \quad (11)$$

*which requires **sufficient variability** in the latent variable $\mathbf{c}$, i.e., the dynamics environments, to be observed.*

*then the parameters $\Phi = (\mathbf{g}, \theta, \zeta)$ are $\sim$-identifiable.*

**Corollary 1:** *When the true dimension $d$ of $\mathbf{c}$ is unknown, setting the dimension of $\hat{\mathbf{c}}$ to $\hat{d} \geq d$ ensures that the original latent variable $\mathbf{c}$ is recovered up to an affine transformation and permutation within a subset of the estimated latent space, while the remaining dimensions encode only noise.*

### 3.2.2. IDENTIFIABILITY OF LATENT STATE $\mathbf{z}_t$

**Theorem 2:** *Suppose there exists an invertible function $\hat{\mathbf{g}}^{-1}$ that $\hat{\mathbf{z}}_t = \hat{\mathbf{g}}^{-1}(\mathbf{x}_t)$. Assume $\hat{\mathbf{z}}_t$ is factorized when conditioned on $\hat{\mathbf{z}}_{t-1}$ and $\mathbf{u}$. Let $\eta_{kt}(\mathbf{u}) = \log p(z_{kt}|\mathbf{z}_{t-1}, \mathbf{u})$ and assume $\eta_{kt}(\mathbf{u})$ is twice differentiable in $z_{kt}$. Let $\nabla^2 \eta_{kt} := \frac{\partial^2 \eta_{kt}}{\partial z_{kt}^2}$ and $\nabla \eta_{kt} := \frac{\partial \eta_{kt}}{\partial z_{kt}}$. If there exists at least $2m + 1$ unique environments $\mathbf{u}_0, \mathbf{u}_1, \ldots, \mathbf{u}_{2m}$ such that the $2m$-dimensional function vectors $\mathbf{s}_{kt}$ and $\mathring{\mathbf{s}}_{kt}$ defined below are linearly independent with $k=1{:}m$, for each value of $\mathbf{z}_t$:*

$$
\begin{aligned}
\mathbf{s}_{kt} &= \left( \nabla^2 \eta_{kt}(\mathbf{u}_1) - \nabla^2 \eta_{kt}(\mathbf{u}_0), \ldots, \right.\\
&\quad \left. \nabla^2 \eta_{kt}(\mathbf{u}_{2m}) - \nabla^2 \eta_{kt}(\mathbf{u}_{2m-1}) \right)^T, \\
\mathring{\mathbf{s}}_{kt} &= \left( \nabla \eta_{kt}(\mathbf{u}_1) - \nabla \eta_{kt}(\mathbf{u}_0), \ldots, \right.\\
&\quad \left. \nabla \eta_{kt}(\mathbf{u}_{2m}) - \nabla \eta_{kt}(\mathbf{u}_{2m-1}) \right)^T,
\end{aligned}
\quad (12)
$$

*which requires **sufficient variability** in the first- and second-order derivatives of $\eta(\mathbf{u})$ to be observed on $\mathbf{z}_t$, then $\hat{\mathbf{z}}_t$ is a permuted invertible component-wise transformation of $\mathbf{z}_t$:*

$$\hat{\mathbf{g}}^{-1}(\mathbf{x}_t) = C \circ P \circ \mathbf{g}^{-1}(\mathbf{x}_t) \Leftrightarrow \hat{\mathbf{z}}_t = C \circ P(\mathbf{z}_t), \quad (13)$$

where $C$ is a component-wise invertible transformation and $P$ is a permutation.

**Corollary 2: (Identifiability of latent dynamics function)** *Given that Theorems 1 and 2 are established, there exist invertible functions $\mathbf{h}$ and $\mathbf{k}$, each a composition of permutations and component-wise transformations, such that*

$$\hat{\mathcal{F}} = \mathbf{h}^{-1} \circ \mathcal{F} \circ \mathbf{k}. \quad (14)$$

*Consequently, the learned transition function $\hat{\mathcal{F}}$ is equivalent to the true transition function $\mathcal{F}$ up to compositions of permutations and component-wise transformations.*

Detailed proofs can be found in Appendix A.

## 4. Practical Implementation of Meta-iLaD

**Variational inference:** To maximize the conditional likelihood in Equations (4)-(7), we approximate the posterior conditional density of $\mathbf{c}$ as $q_\zeta(\mathbf{c}|\mathbf{x}_{0:T}^q \cup \mathcal{D}^s)$, sharing the same networks as its conditional prior. We approximate the posterior density of $\mathbf{z}_0$ with $q_\phi(\mathbf{z}_0|\mathbf{x}_{0:l}^q) = \mathcal{N}(\mathbf{z}_0|\boldsymbol{\mu}_\phi(\mathbf{x}_{0:l}^q), \boldsymbol{\sigma}_\phi(\mathbf{x}_{0:l}^q))$ parameterized by $\phi$ and $l << T$.

Consider data $\mathcal{D}$ that includes time series of $\mathbf{x}_{0:T}$'s generated from $M$ environments: $\mathcal{D} = \{\mathcal{D}_m\}_{m=1}^M$. Variational inference over Equations (4)-(7) involves maximizing the following evidence lower-bound (ELBO) for the conditional likelihood $p_\Phi(\mathbf{x}_{0:T}^q|\mathcal{D}^s)$ of all $\mathbf{x}_{0:T}^q$ and $\mathcal{D}_m$'s:

$$
\begin{aligned}
\mathcal{L}_{\text{ELBO}} = \sum_{\mathcal{D}_m \in \mathcal{D}} \sum_{\mathbf{x}_{0:T}^q \in \mathcal{D}_m} \Big\{ \\
\mathbb{E}_{q_\zeta(\mathbf{c}|\mathbf{x}_{0:T}^q \cup \mathcal{D}^s), q_\phi(\mathbf{z}_0|\mathbf{x}_{0:l}^q)}[\log p_\Theta(\mathbf{x}_{0:T}^q|\mathbf{c}, \mathbf{z}_0)] \\
- \text{KL}(q_\zeta(\mathbf{c}|\mathbf{x}_{0:T}^q \cup \mathcal{D}^s)||p_\zeta(\mathbf{c}|\mathcal{D}^s)) \\
- \text{KL}(q_\phi(\mathbf{z}_0|\mathbf{x}_{0:l}^q)||p(\mathbf{z}_0)) \Big\},
\end{aligned}
\quad (15)
$$

for which the derivation is included in Appendix B. The last two terms in Equation (14) regularize the posterior density of $\mathbf{c}$ and $\mathbf{z}_0$ with their respective priors. The first term in Equation (14) describes the likelihood of each query sample $\mathbf{x}_{0:T}^q \in \mathcal{D}_m$ when its generation is conditioned on the latent environment variable $\mathbf{c}$ inferred from a disjoint set of context samples $\mathcal{D}^s \subset \mathcal{D}_m$, along with $\mathbf{z}_0$ inferred from initial $l + 1$ frames of $\mathbf{x}_{0:T}^q$ itself. In other words, the likelihood of $p_g(\mathbf{x}_t|\mathbf{z}_t)$ at each $t$ for $t \geq 1$ are evaluated at a density $q_\mathcal{F}(\mathbf{z}_t|\mathbf{c})$ following the NODE flow as defined in Equation (6) with the density of $\mathbf{z}_0$ now defined by $q_\phi(\mathbf{z}_0|\mathbf{x}_{0:l}^q)$ as:

$$\log q_\mathcal{F}(\mathbf{z}_t|\mathbf{c}) = \log q_\phi(\mathbf{z}_0|\mathbf{x}_{0:l}^q) - \int_0^t \text{tr}\left(J_{\mathbf{f}_\theta}(\mathbf{z}_\tau, \mathbf{c})\right) d\tau. \quad (16)$$

**Relation to meta-learning:** Equation (14) represents a Bayesian meta-learning objective: $p_\zeta(\mathbf{c}|\mathcal{D}^s)$ and $p_\theta(\mathbf{z}_t|\mathbf{z}_{t-1}, \mathbf{c})$ are feedforward meta-models to extract $\mathbf{c}$ from $k$-shot context samples and to adapt latent dynamics;

the conditional likelihood of all query samples $\mathbf{x}_{0:T}^q$ across all environments accumulates the meta-loss to optimize the meta-models. The practical optimization of Equation (14) involves two encoders: $q_\zeta(\mathbf{c}|\mathbf{x}_{0:T}^q \cup \mathcal{D}^s)$ for the latent environment variable $\mathbf{c}$, and $q_\phi(\mathbf{z}_0|\mathbf{x}_{0:l}^q)$ for initiating the ODE flow-based density $q_\mathcal{F}(\mathbf{z}_t|\mathbf{c})$ as defined in Equation (15). Below we discuss additional regularization strategies to improve the optimization of each encoder.

**Disentanglement of c:** As defined in Equation (4), the latent environment $\mathbf{c}$ is assumed to be conditionally factorized among its $d$ dimensions. To enforce this constraint, we follow FactorVAE (Kim & Mnih, 2018) to minimize its total correlation (TC) by $\mathcal{L}_{\text{TC}} = \mathbb{E}_{q_\zeta}\left[\log \frac{q_\zeta}{\bar{q}_\zeta}\right]$ where $\bar{q}_\zeta := \prod_{j=1}^d q_\zeta(c_j)$, approximated by a discriminator trained to distinguish between samples from $q_\zeta$ and $\bar{q}_\zeta$.

**Warm-up training strategy:** The optimization of $q_\phi(\mathbf{z}_0|\mathbf{x}_{0:l}^q)$ in Equation (14) in theory receives loss signals from the likelihood at all time frames $t{=}0{:}T$ through $q_\mathcal{F}(\mathbf{z}_t|\mathbf{c})$ as defined in Equation (15). In practice, because the calculation of $q_\mathcal{F}(\mathbf{z}_t|\mathbf{c})$ at $t \geq 1$ involves iterative ODE solver through multiple time steps, back propagation of the gradient to $q_\phi$ could be difficult. To mitigate this optimization challenge, we adopt a *warm-up* training strategy to first obtain a reasonable optimization of $q_\phi$ via a *reconstruction* loss (where $\mathbf{x}_{0:T}^q$ is reconstructed from $q_\phi(\mathbf{z}_t|\mathbf{x}_{t:t+l}^q)$), before optimizing the *forecasting* meta-objective (where $\mathbf{x}_{0:T}^q$ is calculated from $q_\mathcal{F}(\mathbf{z}_t|\mathbf{c})$ as defined in Equations (14)-(15)). For transition, we add a regularization term between the two densities $q_\phi(\mathbf{z}_t|\mathbf{x}_{t:t+l}^q)$ and $q_\mathcal{F}(\mathbf{z}_t|\mathbf{c})$ at $t \geq 1$: $\mathcal{L}_{\text{Flow}} = \mathbb{E}_{q_\zeta,q_\phi(\mathbf{z}_0|\mathbf{x}_{0:l}^q)}\left[\sum_{t=1}^T \text{KL}[q_\phi(\mathbf{z}_t|\mathbf{x}_{t:t+l}^q)\|q_\mathcal{F}(\mathbf{z}_t|\mathbf{c})]\right]$.

**Unified loss function:** The two regularization losses $\mathcal{L}_{\text{TC}}$ and $\mathcal{L}_{\text{Flow}}$ are added to $\mathcal{L}_{\text{ELBO}}$ with their respective hyperparamters $\lambda_{\text{TC}}$ and $\lambda_{\text{Flow}}$ as:

$$\mathcal{L}_{\text{Total}} = -\mathcal{L}_{\text{ELBO}} + \lambda_{\text{TC}}\mathcal{L}_{\text{TC}} + \lambda_{\text{Flow}}\mathcal{L}_{\text{Flow}} \qquad (17)$$

where $\lambda_{\text{Flow}}$ is annealed during warm-up. The number of warm-up epochs is determined by the training loss. The effect of these two regularizations is ablated in Section 5.3.

## 5. Experiments & Results

### 5.1. Experiments on Synthetic Physics Systems

**Datasets:** For a comprehensive identifiability assessment, we first considered three benchmark physics systems with known ground truth dynamics: forced damped Pendulum (Takeishi & Kalousis, 2021), CartPole (Song et al., 2023), and Double Pendulum (Wehenkel et al., 2023). For each system, the dynamics environments are characterized by governing physical parameters (*e.g.*, damping coefficient, mass, length), which we randomly sampled to generate image

sequences with varying initial conditions. This controlled setting enabled rigorous quantitative evaluation of identifiability at both the latent state and environment variable levels. Additional data details are provided in Appendix C.1.

**Baselines:** We evaluated Meta-iLaD against two categories of locally-stationary latent dynamics models $\mathcal{F}(\mathbf{z}_{<t}; \mathbf{c})$. For *identifiable* models, we included TDRL (Yao et al., 2022) and our own implementation of IDF (due to absence of public versions) (Hızlı et al., 2025): as previously noted, both baselines leverage predefined environment labels to condition the latent dynamics, with which IDF further established the identifiability of a latent process noise $\mathbf{s}_t$ and $\mathcal{F}$.

For models *without established identifiability*, we included: 1) ODE2VAE which does not infer latent environment variables (Yıldız et al., 2019), and 2) MoNODE which infers environment variables from the query sample itself (Auzina et al., 2023)—note that the subtle difference in inferring $\mathbf{c}$ from context samples (Meta-iLaD) *vs.* the query sample itself (MoNODE) makes a fundamental difference in the definition of a conditional prior of $p(\mathbf{c}|\mathcal{D}_s)$ *vs.* unconditional prior $p(\mathbf{c})$, which is the key to our identifiability results.

**Evaluation scenarios:** We evaluated all models under three progressively challenging settings. First, in **reconstruction**, the full test sequence was observed by the model to assess the recovery of $\mathbf{z}_t$ and $\mathbf{c}$—ODE2VAE by design does not apply in this setting. Second, in **prediction**, the model observed only the first $l$ frames of a test sequence and predicts future trajectories from the estimated $\mathbf{z}_0$ to evaluate whether the learned $\mathcal{F}$ and $\mathbf{c}$ captures the true underlying dynamics function—$\mathbf{c}$ came from context samples in Meta-iLaD, the environment label in IDF, and a previous query sample in MoNODE; TDRL by design does not apply in this setting. Finally, in **OOD generalization**, prediction was performed on dynamics environments with parameters outside the training distribution, TDRL and IDF cannot apply here due to their use of predefined environment labels.

**Metrics:** The performance of all models in the above scenarios were evaluated at both the data and latent space. **Data space** quality was measured by mean squared errors (MSE) between model outputs and ground truth $\mathbf{x}_t$. **Identifiability** quality of latent variables was measured by mean correlation coefficients (MCC) with known ground-truth, adopting the strong-MCC metric in (Khemakhem et al., 2020a;b) a value closer to 1 indicating better identifiability.

**Reconstruction & prediction results:** Table 1 summarizes the full quantitative results on Pendulum and CartPole. Fig. 2 provides MCC visuals on these two datasets for the three identifiable models during reconstruction. Additional results are provided in Appendix C.2.

As shown, for ODE2VAE and MoNODE (without identifiability established), poor MCC($\mathbf{z}_t$) and weaker MSE($\mathbf{x}_t$)

*Table 1.* Quantitative results on Pendulum and CartPole.

| Dataset | Method | Reconstruction | | | Prediction | | | OOD Generalization | | |
|---|---|---|---|---|---|---|---|---|---|---|
| | | MSE ↓ | MCC(z) ↑ | MCC(c) ↑ | MSE ↓ | MCC(z) ↑ | MCC(c) ↑ | MSE ↓ | MCC(z) ↑ | MCC(c) ↑ |
| Pendulum | **Meta-iLaD** | **3.58(0.27)e-1** | **1.00(0.00)** | **0.98(0.01)** | **3.56(0.35)e-1** | **1.00(0.00)** | **0.98(0.01)** | **5.89(0.29)e-1** | **1.00(0.00)** | **0.97(0.00)** |
| | ODE2VAE | / | / | / | 7.26(0.10)e0 | 0.88(0.01) | / | 7.97(0.13)e0 | 0.86(0.01) | |
| | MoNODE | 3.67(0.06)e0 | 0.23(0.06) | 0.24(0.07) | 4.62(0.26)e0 | 0.23(0.06) | 0.30(0.11) | 5.83(0.81)e0 | 0.19(0.07) | 0.26(0.05) |
| | TDRL | 3.98(0.27)e-1 | 0.97(0.02) | 0.14(0.01) | / | / | / | / | / | / |
| | IDF | 3.78(0.34)e-1 | 0.98(0.01) | 0.04(0.01) | 1.26(0.47)e1 | 0.78(0.00) | 0.02(0.01) | / | / | / |
| CartPole | **Meta-iLaD** | **2.68(0.06)e-1** | **0.99(0.00)** | **0.99(0.00)** | **2.69(0.06)e-1** | **0.99(0.00)** | **0.99(0.00)** | **4.92(0.05)e-1** | **0.99(0.00)** | **0.99(0.00)** |
| | ODE2VAE | / | / | / | 5.79(0.15)e0 | 0.89(0.06) | / | 6.81(0.10)e0 | 0.85(0.07) | |
| | MoNODE | 8.57(0.09)e-1 | 0.69(0.07) | 0.06(0.01) | 5.09(0.16)e0 | 0.59(0.17) | 0.20(0.03) | 5.47(0.63)e0 | 0.58(0.15) | 0.25(0.09) |
| | TDRL | 5.93(0.09)e-1 | 0.91(0.01) | 0.16(0.00) | / | / | / | / | / | / |
| | IDF | 3.62(0.03)e-1 | 0.98(0.01) | 0.90(0.00) | 1.52(0.01)e0 | 0.97(0.01) | 0.16(0.04) | / | / | / |

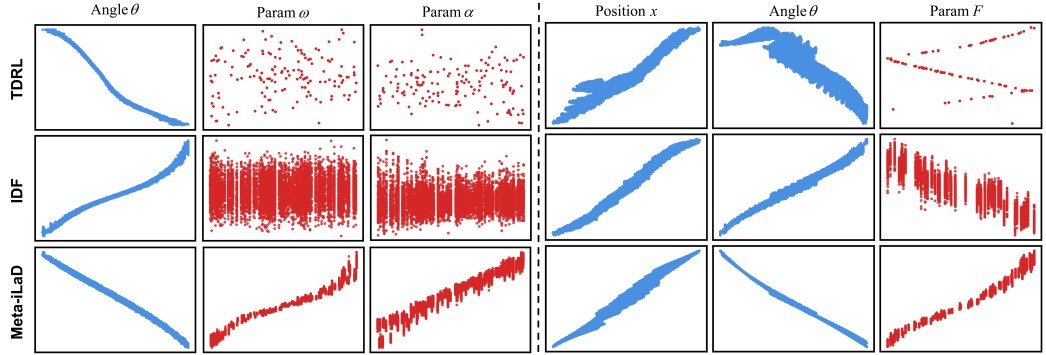

*Figure 2.* Scatter plots of estimated *vs.* true latent states $\mathbf{z}_t$ (blue) and environment variable $\mathbf{c}$ (red). Left: Pendulum; Right: CartPole.

were observed in both reconstruction and prediction settings. Although MoNODE inferred latent environment variable $\mathbf{c}$, MCC($\mathbf{c}$) was low in reconstructions and predictions.

In comparison, the three identifiable models showed much stronger MCC($\mathbf{z}_t$) and thus strong MSE($\mathbf{x}_t$) during reconstruction. Interestingly, while this strong performance was underlined by an equally strong MCC($\mathbf{c}$) in Meta-iLaD, TDRL and IDF delivered either poor or inconsistent MCC($\mathbf{c}$) during reconstruction. Similar performance was observed during prediction, where Meta-iLaD delivered significantly stronger MCC($\mathbf{c}$) than IDF. Unlike the reconstruction setting, this difference in MCC($\mathbf{c}$) directly translated to a significant difference in MCC($\mathbf{z}_t$) and MSE($\mathbf{x}_t$) between the two models. The low MCC($\mathbf{c}$) for IDF suggest that the assumption of process noises in IDF restricted its use to represent more general environment variables, such as parameters of the dynamics equations in these experiments.

**OOD results:** The two identifiable baselines (TDRL and IDF) were not applied to OOD settings due to their use of predefined environment labels. ODE2VAE, while applicable, delivered limited performance potentially due to its non-adaptable latent dynamics. MoNODE with its inference of latent environment variable improved this performance at the data space, but only to a limited extent potentially due to the lack of identifiability. In comparison, Meta-iLaD delivered consistently strong results across all metrics.

### 5.2. Experiments on Real-World CMU Data

**Datasets:** We further evaluated on a subset of the CMU MoCAP dataset consisting of 62 walking sequences from 5 different subjects. Each subject was considered a distinct dynamics environment. Following (Wang et al., 2007), we preprocessed the data to obtain 50-dimensional observation sequences. During testing, models receive the first 75 time steps as input and predict the full 125-step horizon. We used all five subjects for in-distribution (ID) and left one subject out for OOD. Additional data details are in Appendix C.1.

**Baselines & Metrics:** We included baselines capable of prediction. Due to the lack of true latent variables, model performance was measured by MSE($\mathbf{x}_t$) when reconstructing the first 75 time steps and predicting the next 50 steps.

**Results:** As summarized in Table 2, the two identifiable models showed stronger performance in **ID setting**, with Meta-iLaD delivering the most accurate reconstructions and predictions. Fig. 3 provides visual examples of this difference, noting the improvements of Meta-iLaD when predicting over longer horizons. All baseline models showed significant deterioration when applied to the **OOD setting**: the significant drop of performance for IDF, including reconstruction, suggested the limitations of utilizing predefined environment labels. The performance of Meta-iLaD remained the most stable, especially in prediction. Additional results on MoCAP are included in AppendixC.3.

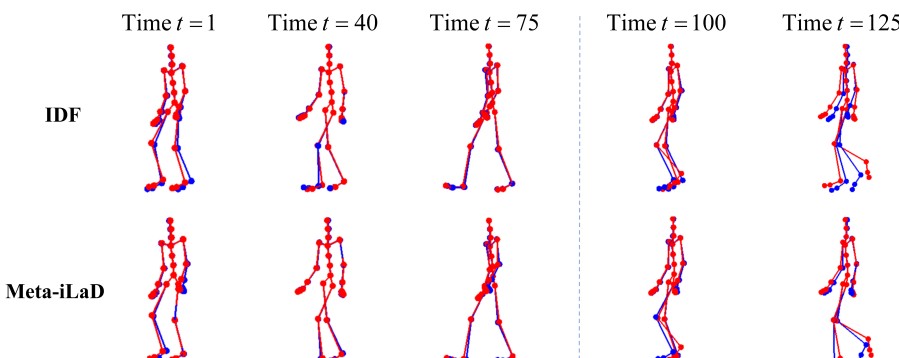

*Figure 3.* Examples of IDF *vs.* Meta-iLaD in reconstruction (t=0:75) and prediction (t=75:125) on an in-distribution subject.

*Table 2.* Reconstruction (0-75) and prediction (75-125) MSEs on MoCAP, tested on in-distribution (ID) and OOD subjects.

| Models | ID | | OOD | |
|---|---|---|---|---|
| | 0-75 | 75-125 | 0-75 | 75-125 |
| ODE2VAE | 2.56(0.00)e4 | 2.55(0.00)e4 | 2.74(0.00)e4 | 2.73(0.00)e4 |
| MONODE | 1.40(0.23)e1 | 3.87(0.46)e1 | 5.34(0.43)e1 | 1.08(0.04)e2 |
| IDF | 6.12(0.42)e0 | 3.21(0.18)e1 | 7.65(1.09)e1 | 1.18(0.06)e2 |
| **Meta-iLaD** | **2.80(0.49)e0** | **1.95(0.11)e1** | **3.29(1.23)e1** | **7.87(1.25)e1** |

*Table 3.* Ablation of conditioning $\mathbf{c}$ on few-shot context samples (representing Meta-iLaD), query sample itself (representing an unconditional prior for $\mathbf{c}$), predefined indices of environments (representing a realization of iVAE in our model), and true environment variables (representing the best scenario).

| | Prediction | | OOD Generalization | |
|---|---|---|---|---|
| | MSE(xt) | MCC(c) | MSE(xt) | MCC(c) |
| Context Samples | 3.49(0.40)e-1 | 0.98(0.00) | 6.16(0.08)e-1 | 0.96(0.00) |
| Query Samples | 5.06(0.50)e-1 | 0.91(0.00) | 8.88(2.75)e-1 | 0.88(0.01) |
| Predefined Index | 3.45(0.22)e-1 | 0.97(0.00) | / | / |
| True Param Value | 3.23(0.10)e-1 | 0.99(0.00) | 5.06(0.20)e-1 | 0.98(0.00) |

## 5.3. Ablation Studies

All ablation experiments were performed on the Pendulum system with ID and OOD environments.

**Meta-Learning via context samples:** A core innovation of Meta-iLaD is conditioning $\mathbf{c}$ on few-shot context samples, rather than predefined labels (*e.g.*, TDRL and IDF) or inferred from the query sample itself (*e.g.*, MoNODE). While the comparison with relevant baselines already validated this design choice, here we further ablated its importance when controlling all other differences. We compared Meta-iLaD with 1) a variant that infers $\mathbf{c}$ from the query sample and 2) a variant that conditions on predefined environment labels $u$—this last variant can be viewed as an realization of iVAE (Khemakhem et al., 2020a) in our model: we considered both true environment parameter values as $u$ as the best-case performance, and discrete labels of $u$ as its practical variant.

As summarized in Table 3, inferring $\mathbf{c}$ from the query samples resulted in limited identifiability, especially in OOD settings. The use of predefined label of environments delivered comparable performance to Meta-iLaD, but could not be generalized to OOD environments. Meta-iLaD delivered consistent performance to the best-case scenario using true environment variables to condition $\mathbf{c}$, across test settings.

**TC regularization & Warm-up:** Table 4 summarizes ablation results on the effect of the two training strategies we used to improve the optimization of Meta-iLaD: $\mathcal{L}_{TC}$ to enforce the conditionally factorized prior of $p_\zeta(\mathbf{c}|\mathcal{D}^s)$, and the warm-up strategy to help with the optimization of

*Table 4.* Ablation of the presence/absence of TC regularization $\mathcal{L}_{TC}$ and warm-up training strategy (tied to $\mathcal{L}_{flow}$.

| $\mathcal{L}_{TC}$ | $\mathcal{L}_{flow}$ | Prediction | | |
|---|---|---|---|---|
| | | MSE(xt) | MCC(zt) | MCC(c) |
| ✗ | ✗ | 3.87(0.30)e0 | 0.20(0.10) | 0.86(0.06) |
| ✓ | ✗ | 3.58(0.27)e0 | 0.26(0.08) | 0.95(0.01) |
| ✗ | ✓ | 4.82(0.22)e-1 | 1.00(0.00) | 0.88(0.02) |
| ✓ | ✓ | 3.49(0.40)e-1 | 1.00(0.00) | 0.98(0.00) |

$q_\phi(\mathbf{z}_t|\mathbf{x}^q_{t:t+l})$. As shown, the presence/absence of $\mathcal{L}_{TC}$ has a significant effect on the MCC($\mathbf{c}$) metrics, while the presence/absence of warm-up has a significant effect on the MCC($\mathbf{z}_t$) metrics, supporting their role as intended.

**Additional results:** In Appendix D, we provide empirical evidence for Corollary 1, for Meta-iLaD's ability to recover the true dimension of $\mathbf{c}$. In Appendix E, we provide real-data application scenarios for Meta-iLaD when knowledge about dynamics environments is not able: based on the assumption of local stationarity, Meta-iLaD can use preceding time windows as context samples to support a query window–this is not possible with any existing identifiable $\mathcal{F}(\mathbf{z}_{<t}; \mathbf{c})$.

## 6. Conclusion and Discussions

We present Meta-iLaD to theoretically and empirically show that 1) conditioning latent dynamics on context samples enabled OOD generalizations, and 2) identifiability of latent environment variable $\mathbf{c}$ and dynamics $\mathcal{F}$, while inconsequen-

tial in reconstructions, has a significant impact on the latent dynamics' forecasting capabilities. **Limitations:** Meta-iLaD requires access to few-shot context samples from each dynamics environment to infer **c**. In many real-world scenarios, obtaining multiple samples from the same environment may be challenging, *e.g.*, collecting multiple patient trajectories under identical clinical conditions. Additionally, the use of NODEs introduces computational overhead, and our identifiability guarantees require sufficient environment diversity which may not always be available in practice.

## Impact Statement

This paper presents theoretical works that prove the simultaneous identifiability of latent dynamics and environments via meta-learning. In the context of scientific modeling, this enhances the interpretability of black-box models, allowing researchers to map learned parameters to physical constants with greater confidence. There are also many potential societal consequences of our work, none of which we feel must be specifically highlighted here.

## Acknowledgements

This work is funded by Award NO: R01HL145590 by National Institutes of Health (NIH) / NATIONAL HEART, LUNG, AND BLOOD INSTITUTE (NHLBI)

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

## A. Proof of Identifiability Theory

### A.1. Proof of Theorem 1

Prior to commencing the proof of Theorem 1, we shall introduce the exponential family distribution and the strong exponential family distribution, along with a pertinent lemma concerning the strong exponential family distribution. These preliminaries establish the requisite foundation for our subsequent proof of Theorem 1.

**Definition 3 (Exponential family)** *A univariate exponential family is a set of distributions whose probability density function can be written as*

$$p(x) = Q(x)Z(\boldsymbol{\theta})e^{\langle \mathbf{T}(x), \boldsymbol{\theta} \rangle} \tag{18}$$

*where* $\mathbf{T} : \mathbb{R} \to \mathbb{R}^k$ *is called the sufficient statistic,* $\boldsymbol{\theta} \in \mathbb{R}^k$ *is the natural parameter,* $Q : \mathbb{R} \to \mathbb{R}$ *the base measure and* $Z(\boldsymbol{\theta})$ *the normalization constant. The dimension* $k \in \mathbb{N} \setminus \{0\}$ *of the parameter is always considered to be minimal, meaning that we can't rewrite the density* $p$ *to have the form Equation 17 with a smaller* $k' < k$*. We call* $k$ *the size of* $p$*.*

**Definition 4 (Strongly exponential distributions)** *We say that an exponential family distribution is strongly exponential if for any subset* $\mathcal{X}$ *of* $\mathbb{R}$ *the following is true:*

$$(\exists \boldsymbol{\theta} \in \mathbb{R}^k \mid \forall x \in \mathcal{X}, \langle \mathbf{T}(x), \boldsymbol{\theta} \rangle = const) \implies (l(\mathcal{X}) = 0 \text{ or } \boldsymbol{\theta} = 0) \tag{19}$$

*where* $l$ *is the Lebesgue measure.*

**Lemma 1** *Consider a strongly exponential distribution of size* $k \geq 2$ *with sufficient statistic* $\mathbf{T}(x) = (T_1(x), \dots, T_k(x))$*. Further assume that* $\mathbf{T}$ *is differentiable almost everywhere. Then there exist* $k$ *distinct values* $x_1$ *to* $x_k$ *such that* $(\mathbf{T}'(x_1), \dots, \mathbf{T}'(x_k))$ *are linearly independent in* $\mathbb{R}^k$*.*

**Proof:** Suppose that for any choice of such $k$ points, the family $(\mathbf{T}'(x_1), \dots, \mathbf{T}'(x_k))$ is never linearly independent. That means that $\mathbf{T}'(\mathbb{R})$ is included in a subspace of $\mathbb{R}^k$ of dimension at most $k-1$. Let $\boldsymbol{\theta}$ a non zero vector that is orthogonal to $\mathbf{T}'(\mathbb{R})$. Then for all $x \in \mathbb{R}$, we have $\langle \mathbf{T}'(x), \boldsymbol{\theta} \rangle = 0$. By integrating we find that $\langle \mathbf{T}(x), \boldsymbol{\theta} \rangle = $ const. Since this is true for all $x \in \mathbb{R}$ and for a $\boldsymbol{\theta} \neq 0$, we conclude that the distribution is not strongly exponential, which contradicts our hypothesis.

**Theorem 1** *Assume that we observe data sampled from a generative model defined according to Equation (7) with parameter* $\Phi = (\mathbf{g}, \theta, \zeta)$ *and latent variable* $\mathbf{c}$*. Assume the following holds:*

    *i The set* $\{\mathbf{x}_{0:T} \in \mathcal{X}^{T+1} | \varphi_\epsilon(\mathbf{x}_{0:T}) = 0\}$ *has measure zero, where* $\varphi_\epsilon$ *is the characteristic function of the density* $p_\epsilon$*.*

    *ii The mixing function* $\mathbf{F}_{\mathbf{g},\theta}$ *is injective.*

    *iii The sufficient statistics* $T_{ij}$ *are differentiable almost everywhere, and* $(T_{ij})_{1 \leq j \leq 2}$ *are linearly independent on any subset of* $\mathcal{X}^{T+1}$ *of measure greater than zero.*

    *iv There exist* $2d + 1$ *distinct points* $\mathbf{u}_0, \dots, \mathbf{u}_{2d}$ *such that the matrix*

$$\mathbf{L} = (\boldsymbol{\lambda}_\zeta(\mathbf{u}_1) - \boldsymbol{\lambda}_\zeta(\mathbf{u}_0), \dots, \boldsymbol{\lambda}_\zeta(\mathbf{u}_{2d}) - \boldsymbol{\lambda}_\zeta(\mathbf{u}_0)) \tag{20}$$

    *of size* $2d \times 2d$ *is invertible.*

*then the parameters* $\Phi = (\mathbf{g}, \theta, \zeta)$ *are* $\sim$*-identifiable.*

**Proof:** Suppose we have two sets of parameters $\Phi = (\mathbf{g}, \theta, \zeta)$ and $\hat{\Phi} = (\hat{\mathbf{g}}, \hat{\theta}, \hat{\zeta})$ such that:

$$p_\Phi(\mathbf{x}_{0:T}|\mathbf{u}) = p_{\hat{\Phi}}(\mathbf{x}_{0:T}|\mathbf{u}) \tag{21}$$

$$\int p_{\mathbf{g},\theta}(\mathbf{x}_{0:T}|\mathbf{c}, \mathbf{z}_0)p_\zeta(\mathbf{c}|\mathbf{u})p(\mathbf{z}_0)d\mathbf{c}d\mathbf{z}_0 = \int p_{\hat{\mathbf{g}},\hat{\theta}}(\mathbf{x}_{0:T}|\mathbf{c}, \mathbf{z}_0)p_{\hat{\zeta}}(\mathbf{c}|\mathbf{u})p(\mathbf{z}_0)d\mathbf{c}d\mathbf{z}_0 \tag{22}$$

To simplify without loss of generality, we rewrite the above equation as:

$$\int p(\mathbf{x}_{0:T}|\mathbf{c}, \mathbf{z}_0)p(\mathbf{c}|\mathbf{u})p(\mathbf{z}_0)d\mathbf{c}d\mathbf{z}_0 = \int p(\mathbf{x}_{0:T}|\hat{\mathbf{c}}, \hat{\mathbf{z}}_0)p(\hat{\mathbf{c}}|\mathbf{u})p(\hat{\mathbf{z}}_0)d\hat{\mathbf{c}}d\hat{\mathbf{z}}_0 \tag{23}$$

$$\int \delta(\mathbf{x}_{0:T} - \mathbf{F}_{\mathbf{g},\theta}(\mathbf{c}, \mathbf{z}_0))p(\mathbf{c}|\mathbf{u})p(\mathbf{z}_0)d\mathbf{c}d\mathbf{z}_0 = \int \delta(\mathbf{x}_{0:T} - \mathbf{F}_{\hat{\mathbf{g}},\hat{\theta}}(\hat{\mathbf{c}}, \hat{\mathbf{z}}_0))p(\hat{\mathbf{c}}|\mathbf{u})p(\hat{\mathbf{z}}_0)d\hat{\mathbf{c}}d\hat{\mathbf{z}}_0 \tag{24}$$

$$p(\mathbf{c}|\mathbf{u})p(\mathbf{z}_0)|\mathbf{J}_{\mathbf{F}_{\mathbf{g},\theta}^{-1}}(\mathbf{x}_{0:T})| = p(\hat{\mathbf{c}}|\mathbf{u})p(\hat{\mathbf{z}}_0)|\mathbf{J}_{\mathbf{F}_{\hat{\mathbf{g}},\hat{\theta}}^{-1}}(\mathbf{x}_{0:T})| \tag{25}$$

Equation 24 can also be derived from Equation 22 with a change of variables.

Now consider two different values of $\mathbf{u}$: $\mathbf{u}_i$, $\mathbf{u}_0$

$$\log p(\mathbf{c}|\mathbf{u}_i)p(\mathbf{z}_0)|\det J_{\mathbf{F}_{\mathbf{g},\theta}^{-1}}(\mathbf{x}_{0:\tau})| - \log p(\mathbf{c}|\mathbf{u}_0)p(\mathbf{z}_0)|\det J_{\mathbf{F}_{\mathbf{g},\theta}^{-1}}(\mathbf{x}_{0:\tau})|$$

$$= \log \frac{Z(\mathbf{u}_0)}{Z(\mathbf{u}_i)} + \langle \mathbf{T}(\mathbf{c}), \boldsymbol{\lambda}(\mathbf{u}_i) - \boldsymbol{\lambda}(\mathbf{u}_0)\rangle = \log \frac{Z(\mathbf{u}_0)}{Z(\mathbf{u}_i)} + \langle \mathbf{T}(\mathbf{c}), \bar{\boldsymbol{\lambda}}(\mathbf{u}_i)\rangle \tag{26}$$

Considering the same for $p(\hat{\mathbf{c}}|\mathbf{u})p(\hat{\mathbf{z}}_0)|\mathbf{J}_{\mathbf{F}_{\hat{\mathbf{g}},\theta}}(\bar{\mathbf{x}}_{0:\tau})|$, we get:

$$\log \frac{Z(\mathbf{u}_0)}{Z(\mathbf{u}_i)} + \langle \mathbf{T}(\mathbf{c}), \bar{\boldsymbol{\lambda}}(\mathbf{u}_i)\rangle = \log \frac{\hat{Z}(\mathbf{u}_0)}{\hat{Z}(\mathbf{u}_i)} + \langle \mathbf{T}(\hat{\mathbf{c}}), \bar{\boldsymbol{\lambda}}(\mathbf{u}_i)\rangle \tag{27}$$

$$\langle \mathbf{T}(\mathbf{c}), \bar{\boldsymbol{\lambda}}(\mathbf{u}_i)\rangle = \langle \mathbf{T}(\hat{\mathbf{c}}), \bar{\boldsymbol{\lambda}}(\mathbf{u}_i)\rangle + \log \frac{\hat{Z}(\mathbf{u}_0)Z(\mathbf{u}_i)}{Z(\mathbf{u}_i)\hat{Z}(\mathbf{u}_0)} = \langle \mathbf{T}(\hat{\mathbf{c}}), \bar{\boldsymbol{\lambda}}(\mathbf{u}_i)\rangle + b_i \tag{28}$$

Based on assumption (iii), for $2d + 1$ distinct points $\mathbf{u}^0, ..., \mathbf{u}^{2d}$, we get

$$\mathbf{L}^T\mathbf{T}(\mathbf{c}) = \hat{\mathbf{L}}^T\mathbf{T}(\hat{\mathbf{c}}) + \mathbf{b} \tag{29}$$

$$\mathbf{T}(\mathbf{c}) = \bar{\mathbf{A}}\mathbf{T}(\hat{\mathbf{c}}) + \bar{\mathbf{b}}, \quad \bar{\mathbf{A}} = \mathbf{L}^{-T}\hat{\mathbf{L}} \text{ and } \bar{\mathbf{b}} = \mathbf{L}^{-T}\mathbf{b} \tag{30}$$

$$\mathbf{T}(\mathbf{c}) = \bar{\mathbf{A}}\mathbf{T}(\hat{\mathbf{c}}) + \bar{\mathbf{b}} \tag{31}$$

Based on assumption (ii) and definition 2, we know $p(\mathbf{c}|\mathbf{u})$ is strongly exponential. Since $J_{\mathbf{T}}(\mathbf{c})$ exists and is an $2d \times d$ matrix of rank $d$, which implies the rank of $\bar{\mathbf{A}}$ is $d$. Based on Lemma 1, for each $i \in [1, \ldots, d]$, define $\mathbf{T}_i(c_i) = (T_{i,1}(c_i), T_{i,2}(c_i))$, there exist 2 points $c_i^1, c_i^2$ that $(\mathbf{T}_i(c_i^1), \mathbf{T}_i(c_i^2))$ are linearly independent. Collect those points into 2 vectors $(\mathbf{c}^1, \mathbf{c}^2)$, and concatenate the 2 Jacobians the matrix $\mathbf{Q} = (\mathbf{J}_{\mathbf{T}}(\mathbf{c}^1), \mathbf{J}_{\mathbf{T}}(\mathbf{c}^2))$. Then the matrix $\mathbf{Q}$ is invertible (through a combination of Lemma 1 and the fact that each component of $T$ univariate). Then define,

$$\hat{\mathbf{Q}} = \left(\mathbf{J}_{\mathbf{T}(\mathbf{F}_{\mathbf{g},\theta}^{-1,\mathbf{c}}\circ\mathbf{F}_{\hat{\mathbf{g}},\theta})}(\mathbf{c}^1), \mathbf{J}_{\mathbf{T}(\mathbf{F}_{\mathbf{g},\theta}^{-1,\mathbf{c}}\circ\mathbf{F}_{\hat{\mathbf{g}},\theta})}(\mathbf{c}^2)\right) \tag{32}$$

we get $\mathbf{Q} = \bar{\mathbf{A}}\hat{\mathbf{Q}}$. The invertibility of $\mathbf{Q}$ implies the invertibility of $\bar{\mathbf{A}}$ and $\hat{\mathbf{Q}}$.

Then since $\mathbf{T}_i(c_i) = (c_i, c_i^2)$, the relationship between the latent spaces becomes

$$\begin{pmatrix} \mathbf{c} \\ \mathbf{c}^2 \end{pmatrix} = \bar{\mathbf{A}} \begin{pmatrix} \hat{\mathbf{c}} \\ \hat{\mathbf{c}}^2 \end{pmatrix} + \bar{\mathbf{b}} \tag{33}$$

where the squaring is applied element-wise. We can write $\bar{\mathbf{A}}$ and $\bar{\mathbf{b}}$ in block matrix form as

$$\bar{\mathbf{A}} = \begin{pmatrix} \mathbf{A}^{(1)} & \mathbf{A}^{(2)} \\ \mathbf{A}^{(3)} & \mathbf{A}^{(4)} \end{pmatrix}, \quad \bar{\mathbf{b}} = \begin{pmatrix} \mathbf{b}^{(1)} \\ \mathbf{b}^{(2)} \end{pmatrix} \tag{34}$$

Then

$$\mathbf{c} = \mathbf{A}^{(1)}\hat{\mathbf{c}} + \mathbf{A}^{(2)}\hat{\mathbf{c}}^2 + \mathbf{b}^{(1)} \tag{35}$$

$$\mathbf{c}^2 = \mathbf{A}^{(3)}\hat{\mathbf{c}} + \mathbf{A}^{(4)}\hat{\mathbf{c}}^2 + \mathbf{b}^{(2)} \tag{36}$$

So we can write for each dimension $i$ of $\mathbf{c}$

$$c_i = \sum_j A_{ij}^{(1)}\hat{c}_j + \sum_j A_{ij}^{(2)}\hat{c}_j^2 + b_i^{(1)} \tag{37}$$

$$c_i^2 = \sum_j A_{ij}^{(3)}\hat{c}_j + \sum_j A_{ij}^{(4)}\hat{c}_j^2 + b_i^{(2)} \tag{38}$$

In order to compare the equations, we square the second term of (36) on the right hand side, involving $\hat{c}_j^2$

$$\left(\sum_j A_{ij}^{(2)}\hat{c}_j^2\right)^2 = \sum_j \left(A_{ij}^{(2)}\right)^2 \hat{c}_j^4 + \sum_{j\neq j'} A_{ij}^{(2)} A_{ij'}^{(2)} \hat{c}_j^2 \hat{c}_{j'}^2 \tag{39}$$

The first term with $\hat{c}_j^4$ matches no term in Equation 37, so we have to set $A_{ij}^{(2)} = 0$ for all $i$ and $j$. This simplifies the earlier equation:

$$c_i = \sum_j A_{ij}^{(1)}\hat{c}_j + b_i^{(1)} \tag{40}$$

Then the square of the first term on the right hand side involves terms with $\hat{c}_j \hat{c}_{j'}$ cross terms:

$$\left(\sum_j A_{ij}^{(1)}\hat{c}_j\right)^2 = \sum_j \left(A_{ij}^{(1)}\right)^2 \hat{c}_j^2 + \sum_{j\neq j'} A_{ij}^{(1)} A_{ij'}^{(1)} \hat{c}_j \hat{c}_{j'} \tag{41}$$

So we have to set $A_{ij}^{(1)} A_{ij'}^{(1)} = 0$ for all $j \neq j'$. This means that the $i$-th row of $\mathbf{A}^{(1)}$ can have at most one nonzero entry. It must also have at least one nonzero entry, since if the row were all zero, a row of $\bar{\mathbf{A}}$ would be all zero (since $\mathbf{A}^{(2)} = 0$), but $\bar{\mathbf{A}}$ has full rank. Hence each row of $\mathbf{A}^{(1)}$ has exactly one nonzero entry and no two rows of $\mathbf{A}^{(1)}$ have their nonzero entries in the same column (since $\bar{\mathbf{A}}$ has full rank). Therefore, we can write:

$$c_i = A_{ij}^{(1)}\hat{c}_j + b_i^{(1)} \tag{42}$$

Since each row of $\mathbf{A}^{(1)}$ has exactly one nonzero entry in distinct columns, we can write $\mathbf{A}^{(1)} = A \cdot P$ where $A = \mathrm{diag}(a_1, \ldots, a_d)$ with $a_i \neq 0$ and $P$ is a permutation matrix. In vector form:

$$\mathbf{c} = A \cdot P(\hat{\mathbf{c}}) + \mathbf{b} \tag{43}$$

where $\mathbf{b} = (b_1^{(1)}, \ldots, b_d^{(1)})^\top$. This establishes the equivalence relation in Definition 2. $\qquad\square$

### A.2. Proof of Corollary 1

**Corollary 1** When the true dimension $d$ of the latent variable $\mathbf{c}$ is unknown, setting the model dimension to $\hat{d} \geq d$ ensures that the decoder recovers the original latent variables $\mathbf{c}$ up to an affine transformation and permutation within a subset of the estimated latent space, while the remaining dimensions encode only noise.

**Proof:** Following the proof of Theorem 1, we obtain the relationship:

$$\mathbf{c} = \mathbf{A}\hat{\mathbf{c}} + \mathbf{b} \tag{44}$$

where $A$ is a $d \times \hat{d}$ matrix and each row of $A$ has exactly one nonzero entry. Therefore, we can write:

$$c_i = A_{ij}\hat{c}_j + b_i \tag{45}$$

This establishes that each generating latent environment variable $c_i$ is linearly related to some latent variable $\hat{c}_j$ of the estimating model. This estimated latent variable is uniquely associated with $c_i$, and any estimated latent variables not associated with a generating latent variable $c_i$ (in the case $\hat{d} > d$) encode no information about the generating latent space.

Consequently, the model has decoded the original latent variables $\mathbf{c}$ up to an affine transformation and permutation as a subset of variables in its estimated latent space and has encoded no information (only noise) into the remaining latent variables. $\qquad\square$

### A.3. Proof of Theorem 2

**Theorem 2** Suppose there exists an invertible function $\hat{\mathbf{g}}^{-1}$ that $\hat{\mathbf{z}}_t = \hat{\mathbf{g}}^{-1}(\mathbf{x}_t)$ and the conditional distribution $p(z_{kt}|\mathbf{z}_{t-1}, \mathbf{u})$ changes across $M$ value of the environment variables $\mathbf{u}$, denoted by $\mathbf{u}_1, \mathbf{u}_2, \ldots, \mathbf{u}_M$. Assume the components of $\hat{\mathbf{z}}_t$ are also mutually independent conditional on $\hat{\mathbf{z}}_{t-1}$ and $\mathbf{u}$. Let $\eta_{kt}(\mathbf{u}) = \log p(z_{kt}|\mathbf{z}_{t-1}, \mathbf{u})$ and assume $\eta_{kt}(\mathbf{u})$ is twice differentiable in $z_{kt}$ and is differentiable in $z_{l,t-1}, l = 1, 2, \ldots, n$. Also let

$$\mathbf{s}_{kt} = \left( \frac{\partial^2 \eta_{kt}(\mathbf{u}_2)}{\partial z_{kt}^2} - \frac{\partial^2 \eta_{kt}(\mathbf{u}_1)}{\partial z_{kt}^2}, \ldots, \frac{\partial^2 \eta_{kt}(\mathbf{u}_M)}{\partial z_{kt}^2} - \frac{\partial^2 \eta_{kt}(\mathbf{u}_{M-1})}{\partial z_{kt}^2} \right)^T \tag{46}$$

$$\mathring{\mathbf{s}}_{kt} = \left( \frac{\partial \eta_{kt}(\mathbf{u}_2)}{\partial z_{kt}} - \frac{\partial \eta_{kt}(\mathbf{u}_1)}{\partial z_{kt}}, \ldots, \frac{\partial \eta_{kt}(\mathbf{u}_M)}{\partial z_{kt}} - \frac{\partial \eta_{kt}(\mathbf{u}_{M-1})}{\partial z_{kt}} \right)^T \tag{47}$$

If for each value of $\mathbf{z}_t$, the $2m$ function vectors $\mathbf{s}_{kt}$ and $\mathring{\mathbf{s}}_{kt}$, with $k = 1, 2, \ldots, m$, are linearly independent, then $\hat{\mathbf{z}}_t$ is a permuted invertible component-wise transformation of $\mathbf{z}_t$, that

$$\hat{\mathbf{g}}^{-1}(\mathbf{x}_t) = C \circ P \circ \mathbf{g}^{-1}(\mathbf{x}_t) \Leftrightarrow \hat{\mathbf{z}}_t = C \circ P(\mathbf{z}_t) \tag{48}$$

where $C$ is a component-wise invertible transformation and $P$ is a permutation.

**Proof:** Using $\mathbf{g}$ and $\hat{\mathbf{g}}$, we can relate the ground-truth and estimated latent states $\mathbf{z}_t$ and $\hat{\mathbf{z}}_t$ to each other:

$$\mathbf{z}_t = \mathbf{g}^{-1}(\mathbf{x}_t) = \mathbf{g}^{-1} \circ \hat{\mathbf{g}}(\hat{\mathbf{z}}_t) = \mathbf{h}(\hat{\mathbf{z}}_t) \tag{49}$$

where $\mathbf{h} = \mathbf{g}^{-1} \circ \hat{\mathbf{g}}$. Using $\mathbf{z}_t = \mathbf{h}(\hat{\mathbf{z}}_t)$, we perform change of variables on the conditional latent density $\log p(\hat{\mathbf{z}}_t|\hat{\mathbf{z}}_{t-1}, \mathbf{u})$ as follows:

$$\log p(\hat{\mathbf{z}}_t|\hat{\mathbf{z}}_{t-1}, \mathbf{u}) = \log p(\mathbf{z}_t|\mathbf{z}_{t-1}, \mathbf{u}) + \log |\mathbf{H}_t| \tag{50}$$

where $\mathbf{H}_t = \mathbf{J}_{\mathbf{h}}(\hat{\mathbf{z}}_t)$ is the Jacobian matrix of $\mathbf{h}$ evaluated at $\hat{\mathbf{z}}_t$. We make use of the fact that the components of $\mathbf{z}_t$ are mutually independent conditional on $\hat{\mathbf{z}}_{t-1}$ and $\mathbf{u}$, then for any $i \neq j$, $\hat{z}_{it}$ and $\hat{z}_{jt}$ are conditionally independent given $\hat{\mathbf{z}}_{t-1} \cup \{\hat{\mathbf{z}}_t \setminus \{\hat{z}_{it}, \hat{z}_{jt}\}\}$, such that:

$$\frac{\partial^2 \log p(\hat{\mathbf{z}}_t|\hat{\mathbf{z}}_{t-1}, \mathbf{u})}{\partial \hat{z}_{it} \partial \hat{z}_{jt}} = 0 \tag{51}$$

Let $\log p(\mathbf{z}_t|\mathbf{z}_{t-1}, \mathbf{u}) = \sum_{k=1}^{m} \eta_{kt}(\mathbf{u})$, $\eta_{kt}(\mathbf{u}) = \log p(z_{kt}|\mathbf{z}_{t-1}, \mathbf{u})$, then

$$\frac{\partial^2 \log p(\hat{\mathbf{z}}_t|\hat{\mathbf{z}}_{t-1}, \mathbf{u})}{\partial \hat{z}_{it} \partial \hat{z}_{jt}} = \sum_{k=1}^{m} \left( \frac{\partial^2 \eta_{kt}(\mathbf{u})}{\partial z_{kt}^2} \mathbf{H}_{kit} \mathbf{H}_{kjt} + \frac{\partial \eta_{kt}(\mathbf{u})}{\partial z_{kt}} \frac{\partial \mathbf{H}_{kit}}{\partial \hat{z}_{jt}} \right) - \frac{\partial^2 \log |\mathbf{H}_t|}{\partial \hat{z}_{it} \partial \hat{z}_{jt}} = 0 \tag{52}$$

Using different values for $\mathbf{u}$ take the difference of this equation across them gives

$$\frac{\partial^2 \log p(\hat{\mathbf{z}}_t|\hat{\mathbf{z}}_{t-1}, \mathbf{u}_{r+1})}{\partial \hat{z}_{it} \partial \hat{z}_{jt}} - \frac{\partial^2 \log p(\hat{\mathbf{z}}_t|\hat{\mathbf{z}}_{t-1}, \mathbf{u}_r)}{\partial \hat{z}_{it} \partial \hat{z}_{jt}}$$
$$= \sum_{k=1}^{m} \left[ \left( \frac{\partial^2 \eta_{kt}(\mathbf{u}_{r+1})}{\partial z_{kt}^2} - \frac{\partial^2 \eta_{kt}(\mathbf{u}_r)}{\partial z_{kt}^2} \right) \mathbf{H}_{kit} \mathbf{H}_{kjt} + \left( \frac{\partial \eta_{kt}(\mathbf{u}_{r+1})}{\partial z_{kt}} - \frac{\partial \eta_{kt}(\mathbf{u}_r)}{\partial z_{kt}} \right) \frac{\partial \mathbf{H}_{kit}}{\partial \hat{z}_{jt}} \right] = 0 \tag{53}$$

Therefore, if $\mathbf{s}_{kt}$ and $\mathring{\mathbf{s}}_{kt}$, for $k = 1, 2, \ldots, m$, are linearly independent, $\mathbf{H}_{kit} \mathbf{H}_{kjt}$ has to be zero for all $k$ and $i \neq j$. That is, in each row of $\mathbf{H}$, there is only one non-zero entry. Since $\mathbf{h}$ is invertible, then $\hat{\mathbf{z}}_t$ must be a permuted invertible component-wise transformation of $\mathbf{z}_t$.

### A.4. Proof of Corollary 2

**Corollary 2** Given that Theorems 1 and 2 are established, there exist two invertible functions $\mathbf{h}$ and $\mathbf{k}$, each being a composition of permutations and component-wise transformations, such that

$$\hat{\mathcal{F}} = \mathbf{h}^{-1} \circ \mathcal{F} \circ \mathbf{k} \tag{54}$$

Consequently, the learned transition function $\hat{\mathcal{F}}$ is equivalent to the true transition function $\mathcal{F}$ up to compositions of permutations and component-wise transformations.

**Proof:** Based on Theorem 1 and 2, we can get

$$\mathbf{z}_t = \mathbf{h}(\hat{\mathbf{z}}_t), \mathbf{c} = \mathbf{d}(\hat{\mathbf{c}}) \tag{55}$$

where both functions $\mathbf{h}$ and $\mathbf{d}$ are invertible and compositions of permutations and component-wise transformations. We define

$$\begin{bmatrix} \mathbf{z}_t \\ \mathbf{c} \end{bmatrix} = \begin{bmatrix} \mathbf{h}(\hat{\mathbf{z}}_t) \\ \mathbf{d}(\hat{\mathbf{c}}) \end{bmatrix} = \mathbf{k} \begin{bmatrix} \hat{\mathbf{z}}_t \\ \hat{\mathbf{c}} \end{bmatrix} \tag{56}$$

then $\mathbf{k}$ must also be an invertible function which is a composition of permutations and component-wise transformations. Then,

$$\hat{\mathbf{z}}_{t+1} = \mathcal{F}\left( \begin{bmatrix} \hat{\mathbf{z}}_t \\ \hat{\mathbf{c}} \end{bmatrix} \right) = \mathbf{h}^{-1} \circ \mathcal{F}\left( \begin{bmatrix} \mathbf{z}_t \\ \mathbf{c} \end{bmatrix} \right) = \mathbf{h}^{-1} \circ \mathcal{F} \circ \mathbf{k} \left( \begin{bmatrix} \hat{\mathbf{z}}_t \\ \hat{\mathbf{c}} \end{bmatrix} \right) \tag{57}$$

So that

$$\hat{\mathcal{F}} = \mathbf{h}^{-1} \circ \mathcal{F} \circ \mathbf{k} \tag{58}$$

Therefore, the learned transition function $\hat{\mathcal{F}}$ is equivalent to the true transition function $\mathcal{F}$ up to compositions of permutations and component-wise transformations.

# B. Derivation Of ELBO

$$
\sum_{\mathcal{D}_m \in \mathcal{D}} \sum_{x_{0:\tau}^q \in \mathcal{D}_m} \log p_\Phi(\mathbf{x}_{0:\tau}^q | \mathcal{D}^s)
$$

$$
= \sum_{\mathcal{D}_m \in \mathcal{D}} \sum_{x_{0:\tau}^q \in \mathcal{D}_m} \log \iint p_{\theta,g}(\mathbf{x}_{0:\tau}^q | \mathbf{c}, \mathbf{z}_0) p_\zeta(\mathbf{c} | \mathcal{D}^s) p(\mathbf{z}_0) d\mathbf{c} d\mathbf{z}_0
$$

$$
= \sum_{\mathcal{D}_m \in \mathcal{D}} \sum_{x_{0:\tau}^q \in \mathcal{D}_m} \log \iint p_{\theta,g}(\mathbf{x}_{0:\tau}^q | \mathbf{c}, \mathbf{z}_0) \frac{p(\mathbf{z}_0)}{q_\phi(\mathbf{z}_0 | \mathbf{x}_{0:l}^q)} q_\phi(\mathbf{z}_0 | \mathbf{x}_{0:l}^q)
$$

$$
\cdot \frac{p_\zeta(\mathbf{c} | \mathcal{D}^s)}{q_\zeta(\mathbf{c} | \mathbf{x}_{0:\tau}^q \cup \mathcal{D}^s)} q_\zeta(\mathbf{c} | \mathbf{x}_{0:\tau}^q \cup \mathcal{D}^s) d\mathbf{c} d\mathbf{z}_0
$$

$$
\geq \sum_{\mathcal{D}_m \in \mathcal{D}} \sum_{x_{0:\tau}^q \in \mathcal{D}_m} \iint \left[ \log p_{\theta,g}(\mathbf{x}_{0:\tau}^q | \mathbf{c}, \mathbf{z}_0) - \log \frac{q_\zeta(\mathbf{c} | \mathbf{x}_{0:\tau}^q \cup \mathcal{D}^s)}{p_\zeta(\mathbf{c} | \mathcal{D}^s)} - \log \frac{q_\phi(\mathbf{z}_0 | \mathbf{x}_{0:l}^q)}{p(\mathbf{z}_0)} \right]
$$

$$
\cdot q_\zeta(\mathbf{c} | \mathbf{x}_{0:\tau}^q \cup \mathcal{D}^s) q_\phi(\mathbf{z}_0 | \mathbf{x}_{0:l}^q) d\mathbf{c} d\mathbf{z}_0
$$

$$
= \sum_{\mathcal{D}_m \in \mathcal{D}} \sum_{x_{0:\tau}^q \in \mathcal{D}_m} \mathbb{E}_{q_\zeta(\mathbf{c} | \mathbf{x}_{0:\tau}^q \cup \mathcal{D}^s), q_\phi(\mathbf{z}_0 | \mathbf{x}_{0:l}^q)} \left[ \log p_{\theta,g}(\mathbf{x}_{0:\tau}^q | \mathbf{c}, \mathbf{z}_0) \right]
$$

$$
- \mathrm{KL} \left[ q_\zeta(\mathbf{c} | \mathbf{x}_{0:\tau}^q \cup \mathcal{D}^s) \| p_\zeta(\mathbf{c} | \mathcal{D}^s) \right] - \mathrm{KL} \left[ q_\phi(\mathbf{z}_0 | \mathbf{x}_{0:l}^q) \| p(\mathbf{z}_0) \right]
$$

## C. Additional Details for Experiments in Section 5.1 & 5.2

### C.1. Data descriptions

For each dataset, we generate about 10k samples for both training and testing. A general overview of the datastes is as follows and their schematic/examples in Fig. 4

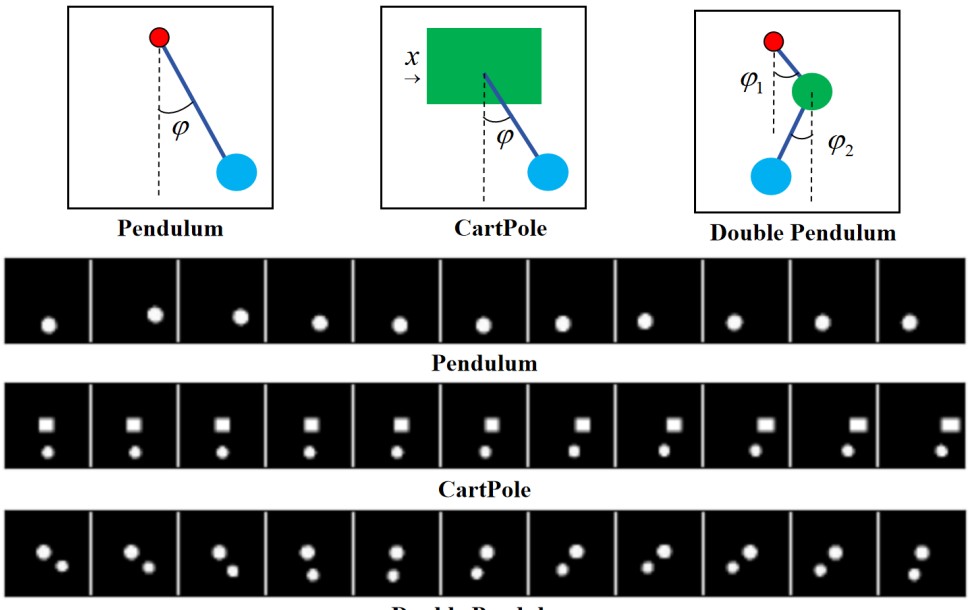

*Figure 4.* The general overview of the datasets

### C.1.1. FORCED DAMPED PENDULUM

$$\frac{d^2\varphi}{dt^2} = -\omega^2 \sin\varphi - \alpha\frac{d\varphi}{dt} + A\cos(2\pi t) \tag{59}$$

We randomly sample $\varphi(t=0) \in [-1.0, 1.0]$, $\dot{\varphi}(t=0) \in [-0.5, 0.5]$ , $\omega \in [2.5, 3.5]$ and $\alpha \in [0.5, 0.8]$, and fix $A = 3.0$ to generate data. For OOD, we consider $\omega \in [2.0, 2.5]$ and $\alpha \in [0.8, 1.0]$

### C.1.2. CARTPOLE

$$\begin{bmatrix} \ddot{x} \\ \ddot{\varphi} \end{bmatrix} = \begin{bmatrix} \frac{-g\sin\varphi\cos\varphi-(F+l\dot{\varphi}^2\sin\varphi))}{\cos^2\varphi-(\tilde{m}+1)l} \\ \frac{(\tilde{m}+1)g\sin\varphi+\cos\varphi(F+l\dot{\varphi}^2\sin\varphi)}{l\cos^2\varphi-(\tilde{m}+1)l} \end{bmatrix} \tag{60}$$

We randomly sample $x(t = 0) \in [-1.0, 1.0]$, $\varphi(t = 0) \in [-1.0, 1.0]$ and $F \in [-2.5, 2.5]$, then fix $\dot{x}(t=0) = 0$, $\dot{\varphi}(t=0) = 0$, $l = 3.0$, $\tilde{m} = 5.0(m_2 = 1.0)$, $g = 9.8$ to generate data. For OOD, we consider $F \in [-4.0, -2.5] \cup [2.5, 4.0]$.

### C.1.3. DOUBLE PENDULUM

$$\begin{bmatrix} \ddot{\varphi}_1 \\ \ddot{\varphi}_2 \end{bmatrix} = \begin{bmatrix} \frac{-g(\tilde{m}+1)\sin\varphi_1-g\sin(\theta_1-2\theta_2)-2\sin(\varphi_1-\varphi_2)(\dot{\varphi}_2^2 l_2+\dot{\varphi}_1^2 l_1\cos(\varphi_1-\varphi_2))}{l_1(2\tilde{m}+1-\cos(2\theta_1-2\theta_2))} \\ \frac{2\sin(\varphi_1-\varphi_2)(\dot{\varphi}_1^2 l_1(\tilde{m}+1)+g(\tilde{m}+1)\cos\varphi_1+\dot{\varphi}_2^2 l_2\cos(\varphi_1-\varphi_2))}{l_2(2\tilde{m}+1-\cos(2\theta_1-2\theta_2))} \end{bmatrix} \tag{61}$$

We randomly sample $\varphi_1(t = 0) \in [-1.0, 1.0]$, $\varphi_2(t = 0) \in [-1.0, 1.0]$, $g \in [5.0, 10.0]$ and , then fix $\dot{\varphi}_1(t = 0) = 0$, $\tilde{m} = 2.0$, $\dot{\varphi}_2(t = 0) = 0$, $l_1 = 1.0$ and $l_2 = 2.5$ to generate data. For OOD, we consider $g \in [10.0, 12.5]$

C.1.4. M O C A P

The walking sequences we consider are `35_33.amc`, `39_08.amc`, `16_16.amc`, `35_06.amc`, `07_02.amc`, `35_10.amc`, `35_14.amc`, `07_06.amc`, `35_13.amc`, `39_06.amc`, `16_21.amc`, `39_07.amc`, `08_01.amc`, `35_31.amc`, `07_01.amc`, `35_07.amc`, `08_10.amc`, `16_15.amc`, `39_01.amc`, `08_02.amc`, `35_16.amc`, `39_14.amc`, `08_03.amc`, `39_13.amc`, `07_09.amc`, `16_58.amc`, `39_04.amc`, `16_31.amc`, `35_32.amc`, `35_34.amc`, `16_22.amc`, `35_08.amc`, `35_01.amc`, `35_12.amc`, `38_02.amc`, `39_03.amc`, `35_11.amc`, `39_02.amc`, `39_10.amc`, `16_47.amc`, `07_03.amc`, `08_06.amc`, `39_12.amc`, `07_08.amc`, `35_29.amc`, `07_10.amc`, `38_01.amc`, `35_09.amc`, `35_05.amc`, `35_02.amc`, `39_05.amc`, `35_28.amc`, `35_04.amc`, `08_08.amc`, `08_09.amc`, `35_03.amc`, `39_09.amc`, `16_32.amc`, `35_30.amc`, `07_07.amc`, `35_15.amc`, `07_11.amc`. The number of training, validation, and test samples in MOCAP-Multiple (ID) and MOCAP-SHIFT (OOD) splits are 52-5-5 and 49-5-8, respectively. Since sequences are already dense, we skip every other data point. For ease of implementation, we take the last 250 time points, leading to sequences of length $T = 125$.

## C.2. Additional Experimental Results on Synthetic Physics Systems

### C.2.1. EXPERIMENTAL RESULTS ON DOUBLE PENDULUM SYSTEMS

We provide additional results on the Double Pendulum system with two-dimensional latent state $\mathbf{z}_t = [\theta_1, \theta_2]$ (angles of the two pendulum arms) and environment variable $G$ (gravitational constant).

Table 5 shows that Meta-iLaD achieves consistently strong performance across all settings with near-perfect MCC scores. In contrast, IDF shows significant degradation in prediction (MSE increases from 8.04e-1 to 1.61e0) with poor MCC(c)=0.08, indicating failure to identify the environment variable. Figure 5 visualizes this through scatter plots: Meta-iLaD (bottom row) exhibits clear linear relationships for both angles and parameter $G$, while IDF (top row) achieves good identifiability for latent angles but fails to recover $G$ (random noise pattern). This directly explains IDF's poor prediction performance—without properly identifying $G$, the learned dynamics function cannot accurately forecast future trajectories.

*Table 5.* Quantitative results on Double Pendulum.

| Double Pendulum | Reconstruction | | | Prediction | | | OOD Generalization | | |
|---|---|---|---|---|---|---|---|---|---|
| | MSE ↓ | MCC(z) ↑ | MCC(c) ↑ | MSE ↓ | MCC(z) ↑ | MCC(c) ↑ | MSE ↓ | MCC(z) ↑ | MCC(c) ↑ |
| **Meta-iLaD** | **6.01(0.03)e-1** | **0.98(0.01)** | **0.98(0.01)** | **6.06(0.02)e-1** | **0.98(0.01)** | **0.98(0.01)** | **9.18(0.02)e-1** | **0.98(0.01)** | **0.98(0.01)** |
| **ODE2VAE** | / | / | / | 8.88(0.07)e0 | 0.60(0.03) | / | 9.94(0.38)e0 | 0.60(0.07) | / |
| **MoNODE** | 1.51(0.24)e0 | 0.85(0.01) | 0.02(0.00) | 5.43(0.16)e0 | 0.70(0.02) | 0.12(0.01) | 6.54(0.30)e0 | 0.75(0.07) | 0.08(0.06) |
| **TDRL** | 8.41(0.04)e-1 | 0.90(0.04) | 0.13(0.02) | / | / | / | / | / | / |
| **IDF** | 8.04(0.03)e-1 | 0.94(0.00) | 0.04(0.02) | 1.61(0.18)e0 | 0.94(0.00) | 0.08(0.04) | / | / | / |

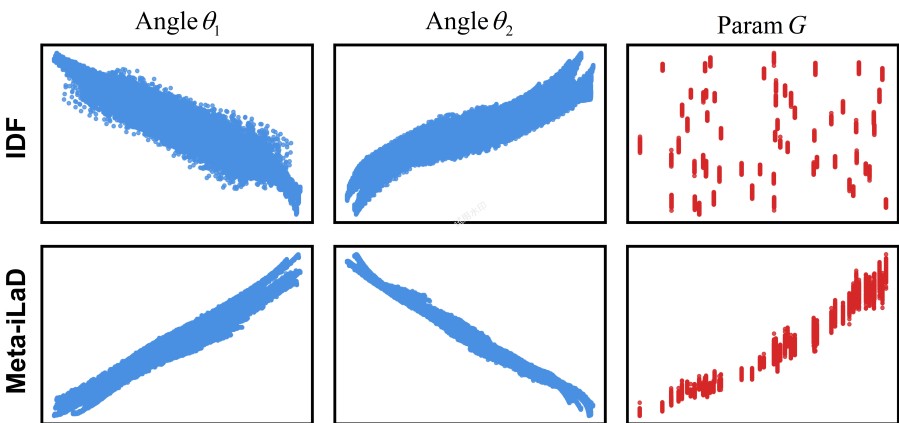

*Figure 5.* Scatter plots of estimated versus true values for latent angles $\theta_1, \theta_2$ (blue) and environment parameter $G$ (red) on Double Pendulum during reconstruction. Meta-iLaD successfully identifies both latent states and environment variable (clear linear relationships), while IDF identifies latent states but fails to recover $G$ (noise pattern).

### C.2.2. OOD GENERALIZATION ON CARTPOLE

Figure 6 demonstrates Meta-iLaD's ability to identify environment parameters in out-of-distribution regions on the CartPole dataset. The model was trained on in-distribution parameter range $[-2.5, 2.5]$ (blue points) and tested on OOD ranges $[-4.0, -2.5] \cup [2.5, 4.0]$ (red points). The scatter plot shows that Meta-iLaD maintains a clear linear relationship between estimated and true parameters across both ID and OOD regions, indicating successful identifiability beyond the training distribution. This ability to correctly identify environment variables in OOD regions directly explains Meta-iLaD's strong OOD generalization performance reported in Table 1.

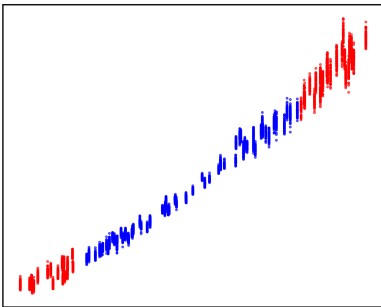

*Figure 6.* Scatter plot of estimated versus true environment parameters on CartPole. Blue points represent in-distribution range $[-2.5, 2.5]$, red points represent OOD ranges $[-4.0, -2.5] \cup [2.5, 4.0]$. Meta-iLaD maintains identifiability across both ID and OOD regions.

### C.3. Additional Experimental Results on Real-World CMU Data

Figure 7 presents qualitative comparisons of trajectory predictions between IDF and Meta-iLaD on the CMU MoCAP dataset. We visualize predictions across six representative joint angles (Joint 3, 13, 27, 44, 46, 47) for an in-distribution test subject. The gray shaded region indicates the test-time input range (frames 0-75), while the unshaded region shows the prediction horizon (frames 76-125).

As shown in the figure, Meta-iLaD (red dash-dot line) consistently tracks the ground truth trajectory (black solid line) more accurately than IDF (blue dashed line) across all joints. The performance gap becomes particularly evident in the prediction phase ($t > 75$), where IDF predictions exhibit significant drift and fail to capture the periodic patterns of the walking motion. In contrast, Meta-iLaD maintains better alignment with the true dynamics, successfully predicting the amplitude and phase of joint movements. These qualitative results align with the quantitative findings in Table 2, demonstrating that identifiability of the environment variable $\mathbf{c}$ and dynamics function $F$ is critical for accurate long-term forecasting.

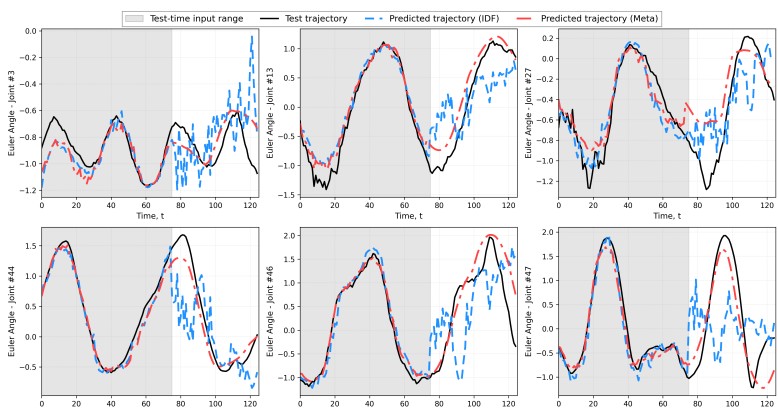

*Figure 7.* Test trajectory predictions for IDF and Meta-iLaD methods. At test time, first 75 time points are given as input, and the prediction horizon is 50 steps (frames 76-125). The figure shows comparisons across multiple joints (Joint 3, 13, 27, 44, 46, 47), where the gray shaded region indicates the input range, the black solid line represents the ground truth trajectory, the blue dashed line shows IDF predictions, and the red dash-dot line shows Meta-iLaD predictions.

## D. Empirical Validation of Corollary 1

Corollary 1 states that when the true dimension $d$ of the latent environment variable $\mathbf{c}$ is unknown, setting the estimated dimension $\hat{d} \geq d$ ensures that the original latent variable $\mathbf{c}$ is recovered up to an affine transformation and permutation within a subset of the estimated latent space, while the remaining dimensions encode only noise.

To validate this corollary, we conducted experiments on the CartPole dataset where the true environment variable $\mathbf{c}$ is one-dimensional ($d = 1$). We trained Meta-iLaD with varying estimated dimensions $\hat{d} \in \{2, 3, 4\}$ and evaluated the learned latent environment dimensions by computing their Mean Correlation Coefficient (MCC) with the true environment parameter. For visualization, we sorted the estimated dimensions $\{\hat{c}_1, \hat{c}_2, \ldots, \hat{c}_{\hat{d}}\}$ in descending order of their MCC values.

**Results:** As shown in Figure 8, across all configurations, only the top-ranked dimension $\hat{c}_1$ (with highest MCC) exhibits a clear monotonic relationship with the true parameter, successfully achieving identifiability. All remaining dimensions $\{\hat{c}_2, \hat{c}_3, \ldots, \hat{c}_{\hat{d}}\}$ display random noise-like patterns with no systematic structure:

- $\hat{d} = 2$: $\hat{c}_1$ shows strong negative linear correlation with true $\mathbf{c}$; $\hat{c}_2$ contains only noise.

- $\hat{d} = 3$: $\hat{c}_1$ captures the true parameter; $\hat{c}_2, \hat{c}_3$ are noisy.

- $\hat{d} = 4$: $\hat{c}_1$ remains informative; $\hat{c}_2, \hat{c}_3, \hat{c}_4$ encode noise.

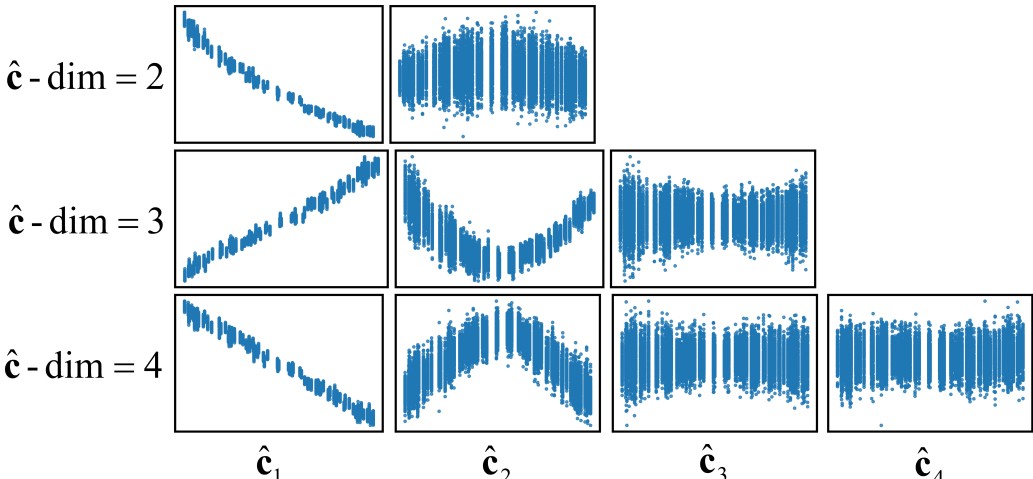

*Figure 8.* Scatter plots of estimated latent environment dimensions $\{\hat{c}_1, \hat{c}_2, \hat{c}_3, \hat{c}_4\}$ (sorted by MCC in descending order) versus the true environment parameter on CartPole dataset. Rows correspond to different estimated dimensions $\hat{d} \in \{2, 3, 4\}$ with true dimension $d = 1$. Only $\hat{c}_1$ exhibits identifiable relationship with the true parameter (clear monotonic trend), while all remaining dimensions contain only noise, confirming Corollary 1.

**Conclusion:** These results empirically confirm Corollary 1: overestimating the dimensionality of $\mathbf{c}$ does not compromise identifiability. The model successfully recovers the true one-dimensional structure in exactly one dimension (the highest-MCC dimension), while all excess dimensions naturally collapse to noise, demonstrating that Meta-iLaD's identifiability guarantees remain robust under dimension misspecification.

# E. Beyond Known Dynamics Environments

Finally, we ventured into settings where there are no prior knowledge about the underlying dynamics environments. Based on the assumption of local stationarity (dynamics environments do not change rapidly over time), Meta-iLaD used $k$-shot preceding time windows as context samples for the current query window to predict.

**Baselines:** Because identifiable latent dynamics utilizing predefined environment labels $u$ does not apply here, we considered NCTRL (Song et al., 2023) as representatives of identifiable latent dynamics $\mathcal{F}(\mathbf{z}_{<t}; \mathbf{c}_t)$ where $\mathbf{c}_t$ is assumed to represent the latent states of a Markov process.

**Real data & results:** We considered a real double-pendulum dataset (Wehenkel et al., 2023) with unknown dynamics environments. NCTRL was initialized with an arbitrary number of hidden states $= 10$. Since ground truth for $\mathbf{z}_t$'s was unknown, we followed (Willetts & Paige, 2021) to calculate MCC between $\mathbf{z}_t$'s obtained between multiple runs. As shown in Fig. 9B, NCTRL demonstrated a strong MSE performance for reconstructing $\mathbf{x}_t$'s, although less successful with the identifiability for the latent $\mathbf{z}_t$'s. In comparison, even tested in the more challenging setting of forecasting, Meta-iLaD was able to achieve stronger results in both metrics.

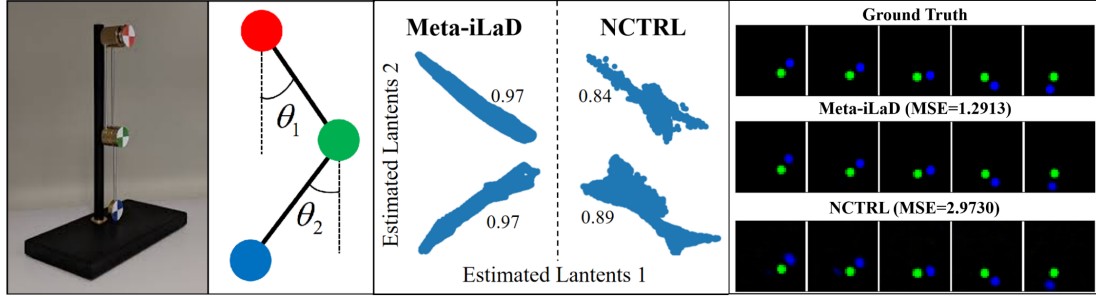

*Figure 9.* Results on real double pendulum data (left), including MCC on latent $\mathbf{z}_t$'s (middle) and examples on reconstructing (NCTRL) or forecasting (Meta-iLaD) $\mathbf{x}_t$'s (right).

This set of exploratory experiments demonstrated the real-world versatility of Meta-iLaD in problems where the underlying latent dynamics environments are not expected to rapidly switch. Expectedly, incorrect context-query pairing near the near the unknown boundary of environment switch will likely impact the performance of Meta-iLaD.

# F. Model Architecture and Implementation Details

## F.1. E.1. Model Architecture

Table 6 summarizes the neural network architectures for the main components of Meta-iLaD.

*Table 6.* Architecture details of Meta-iLaD components.

| VAE Encoder (for $\mathbf{z}_t$) | VAE Decoder (for $\mathbf{z}_t$) |
|---|---|
| Input: $\mathbb{R}^{L \times H \times W}$, window size $k = 3$ | Input: $\mathbb{R}^{d_z/2}$ |
| Conv2d($k$, 32, 4, stride=2) + ReLU | FC($d_z/2$, $128 \times \frac{H}{8} \times \frac{W}{8}$), Unflatten |
| Conv2d(32, 64, 4, stride=2) + ReLU | ConvTranspose2d(128, 64, 4, stride=2) + ReLU |
| Conv2d(64, 128, 4, stride=2) + ReLU | ConvTranspose2d(64, 32, 4, stride=2) + ReLU |
| Flatten | ConvTranspose2d(32, 1, 4, stride=2) |
| FC to $\mu_z$, $\sigma_z$ (dim=$d_z$) | Output: $\mathbb{R}^{H \times W}$ |
| **Environment Encoder (for c)** | **Dynamics Function $F$** |
| Spatial: Conv2d(1, 16, 3, stride=2) + ReLU + BN | Input: $[\mathbf{z}_t, t, \mathbf{c}] \in \mathbb{R}^{d_z+1+d_c}$ |
|        Conv2d(16, 32, 3, stride=2) + ReLU + BN | MLP: FC(5, 64) + SiLU |
|        Conv2d(32, 64, 3, stride=2) + ReLU + BN |     FC(64, 64) + SiLU ($\times$3 layers) |
| Temporal: Conv1d(feat, 256, 5) + ReLU + BN |     FC(64, 1) |
|        Conv1d(256, 256, 5, stride=2) + ReLU + BN | Output: acceleration $\in \mathbb{R}$ |
|        AdaptiveAvgPool1d(8) | Solved via ODE solver (RK4) |
| FC(256$\times$8, 128) + ReLU + Dropout(0.3) | |
| FC(128, 32) + ReLU + Dropout(0.2) | **Discriminator (for TC loss)** |
| FC to $\mu_c$, $\sigma_c$ (dim=$d_c$) | FC($d_c$, 128) + LeakyReLU(0.2) |
| | FC(128, 128) + LeakyReLU(0.2) ($\times$2) |
| | FC(128, 2) |

## F.2. E.2. Implementation Details

**Training configuration.** All models were trained using Adam optimizer with learning rate $1 \times 10^{-3}$ for the main network and $5 \times 10^{-4}$ for the discriminator.

**Loss hyperparameters.** The total loss is $\mathcal{L}_{\text{Total}} = -\mathcal{L}_{\text{ELBO}} + \lambda_{\text{TC}}\mathcal{L}_{\text{TC}} + \lambda_{\text{Flow}}\mathcal{L}_{\text{Flow}}$. We set $\lambda_{\text{TC}} = 5.0$ following FactorVAE. For $\lambda_{\text{Flow}}$, we used a warm-up annealing schedule: starting at 0, linearly increasing to 0.1 over 50 epochs.

**Hardware.** All experiments were conducted on a single NVIDIA RTX 3090 GPU with 24GB memory. Training time ranged from 1-2 hours for synthetic physics system datasets.

