# OpenReview forum: "Meta-iLaD: Identifiable Latent Dynamics via Meta-Learning of Dynamics Environments"
_ICML.cc/2026/Conference — ICML 2026 regular_

### Official Review · Reviewer_soYL · 2026-02-18

**Soundness:** 2
**Presentation:** 1
**Significance:** 2
**Originality:** 2
**Overall Recommendation:** 2
**Confidence:** 4

**Summary:**

This paper tries to solve two limitations including reliance on predefined labels and limited understanding of the identifiability. A novel identifiable latent dynamics framework is proposed by meta-learning. Experiments on synthetic and real data show empirical evidence against claims in this research.

**Compliance With Llm Reviewing Policy:**

Affirmed.

**Key Questions For Authors:**

Please see weaknesses.

**Limitations:**

Yes.

**Strengths And Weaknesses:**

Strengths:

(1) The methodology seems approprite and details are sufficient.

(2) The motivation of this research is beneficial to the community and may be used in diverse application areas.

Weaknesses:

(1) High relyment on assumptations. The identifiability proofs rely on several strong assumptions such as linear independence of sufficient statistics. While in the nonlinear ICA literature, these assumptions are difficult to satisfy, which may limit the applicability.

(2) There are lack of failure cases. The discussion on the sensitivity of shots, the impact of noisy context or performance degradation when assumptions are violated are necessary.

(3) Computational cost is not reported. As the high dimensional systems may endanger the applicability of this kind of model, it is necessary to report cost and speed.

(4) There are too many equations in Abstract, and some paragraphs in Introduction. This may affect the reading accuracy.

(5) Implementation details are severely lacking, being very limited in F.2.

(6) It is difficult to understand the equation in Line 581.

(7) Serious writing issues. E.g., in Abstract, Line23-24, the symbol '—' seems unreasonable and Line28, 'Meta-Lad' should be 'Meta-LaD'.

---

> ### Author Rebuttal · Authors · 2026-03-31
>
> Thank you for your review. We address each point below.
>
> - **Theoretical assumptions:** The assumptions in our work are standard in the identifiability literature and shared by closely related works such as iVAE and TDRL. In particular, the linear independence of sufficient statistics is not a strong assumption in our setting — since we adopt a Gaussian prior, the sufficient statistics take the form $T_{ij} = (c, c^2)$, whose linear independence holds trivially and requires no additional conditions to verify. Assumption 1 concerns the noise distribution and is satisfied by standard Gaussian noise. Assumption 2 concerns the injectivity of the mixing function, which is readily satisfied by construction in synthetic datasets and is also a standard assumption in the nonlinear ICA literature. Assumption 4 requires sufficient variability across environments, with a theoretical minimum of only $2d+1$ distinct environments. However, this bound assumes idealized conditions that are difficult to achieve in practice — in practice, a larger number of environments is beneficial for obtaining reliable identifiability results, as also evidenced by our experiments on stress-testing this assumption (please see details in the next response).
>
> - **Sensitivity of $k$ and the $2d+1$ environment requirement:** We provide additional experiments addressing both points in Figures 1 and 2 of the [anonymous link](https://anonymous.4open.science/api/repo/ICML26-B224/file/ICML_Rebuttal.pdf?v=6a0a4861). For the sensitivity of $k$, the value was set to 9 in all reported experiments. As $k$ decreases, identifiability of $\mathbf{c}$ degrades gradually, yet even at $k=1$ a meaningful identifiable structure is retained, suggesting that a single context sample already provides useful information about the environment.
>
> | | $k=0$ | $k=1$ | $k=3$ | $k=5$ | $k=7$ | $k=9$ |
> |---|---|---|---|---|---|---|
> | MCC($z$) | 0.9916 | 0.9950 | 0.9986 | 0.9982 | 0.9975 | 0.9985 |
> | MCC($c$) | 0.8918 | 0.9331 | 0.9427 | 0.9528 | 0.9734 | 0.9779 |
>
> For the $2d+1$ environment requirement, MCC($\mathbf{c}$) improves progressively as the number of environments increases. While Theorem 1 establishes $2d+1$ as the theoretical minimum, this bound assumes idealized conditions (e.g., exact distribution matching, infinite data) that are difficult to satisfy in practice. In practice, a substantially larger number of environments is needed to achieve reliable identifiability of $\mathbf{c}$, as reflected in our results.
>
> | Env num ($d=2$) | 5 | 10 | 50 | 100 |
> |---|---|---|---|---|
> | MCC($z$) | 0.9993 | 0.9976 | 0.9972 | 0.9989 |
> | MCC($c$) | 0.7446 | 0.9317 | 0.9428 | 0.9782 |
> - **Computational cost:** We have briefly reported computational cost in Appendix F. Compared to IDF, Meta-iLaD requires additional computation for the continuous normalizing flow used to evaluate the density $q_F(z_t | \mathbf{c})$, which approximately doubles the training time relative to IDF. We will report more detailed figures in the revision.
>
> - **Equations in Abstract and Introduction:** The symbols and equations in these sections are intentionally kept simple, serving primarily to clearly convey the goals of our work and precisely position our contributions within the existing literature. We believe this aids rather than hinders readability, and respectfully disagree that they negatively affect reading accuracy.
>
> - **Implementation details:** Network architectures and hyperparameter choices are outlined in Appendix F, with the core methodology described in Section 4. We will further supplement these details in the revision.
>
> - **Equation at Line 581:** The text `[label=i]` at Line 581 is a LaTeX formatting artifact from the enumerated list environment and is not part of the theorem or its proof. It does not affect the correctness of any theoretical result and will be corrected in the revision.
>
> - **Writing issues:** The dash symbol '—' at Lines 23–24 is used for parenthetical clarification and is standard English punctuation. We acknowledge the typo 'Meta-iLad' at Line 28, which should read 'Meta-iLaD'; we thank the reviewer for the careful reading and will correct it in the revision.

---

### Official Review · Reviewer_1Jb2 · 2026-03-11

**Soundness:** 4
**Presentation:** 2
**Significance:** 3
**Originality:** 2
**Overall Recommendation:** 4
**Confidence:** 3

**Summary:**

The authors propose a method for learning dynamical systems that allows for the joint identifiability of latent state, transition function, and environment variable. In a meta-learning framework (similar to a continuous mixture-of-experts), the environment variables are to inferred from a few sample sequences and the dynamics is modelled through conditional transition model (deterministic neural ODE).  Specifically, explicit learning of environment variables should enable generalization to unseen environments. Theoretical results show that, under certain conditions, the parameters of such models are unique up to scaling, permutations, and translations of the environment labels.
The methodology is validated using multiple established data sets (pendulum, double pendulum, cartpole, CMU mocap) and compared to existing methods (ODE2VAE, MoNODE, TDRL, IDF).

**Compliance With Llm Reviewing Policy:**

Affirmed.

**Key Questions For Authors:**

The authors write that the goal of the reconstruction is “to assess the recovery of $z_t$.” As I understand it, the physical states (e.g., the angle of the pendulum) are already passed to the model as observations. The experiments therefore do not validate “state discovery” but rather “representation alignment.”

Are the conditions of Theorem 1 satisfied by the experiments? What are the sufficient statistics on $c$ here, and are they linearly independent? Empirical datasets often vary only weakly such that not all latent dimensions are excited.

The theory implies an infinite amount of data, since $p_\Phi(x_{0:T}\vert u) = p_\hat{Phi}(x_{0:T}\vert u)$ is taken as the starting point for identifiability. Can quantitative conclusions also be drawn for the experimentally relevant case of a finite amount of data?

Why was gravitational acceleration taken as a varying environment parameter for the pendulum and not mass or length? In practice, it is these  rather than gravity that change.

**Strengths And Weaknesses:**

Strengths:

The identifiability of latent states and parameters is a key problem in reconstructing and predicting dynamic systems. In addition, systems with different parameter configurations (“environments”) are discussed here, which makes the treatment of identifiability more challenging.
The authors provide both theoretical and practical results based on benchmark data, which indicate both correct prediction and accurate identification of environment labels.

Particularly useful for working with empirical data is the corollary, which ensures that even if the unknown dimension of $c$ is chosen too high, the model will still learn the true environmental variables, with additional dimensions encoding only noise.

Weaknesses:

The figures convey the results only very vaguely in some cases. For example, why are the actual and predicted environment variables not compared (Figure 2, Figure 5)? I also find it hindering to assess performance when none of the plots have x and y scales. Furthermore, I do not fully understand the connection between the latent states and the environment parameters: If the latent states all belong to one environment (one red dot), which one is it?
I think a clearer description of the figures and detailed captions would greatly improve the paper.

In my view, the experimental section only addresses the theoretical results to a limited extent. Mathematically, the paper deals with the identifiability of latent dynamicaö systems. The experiments, on the other hand, prove that meaningful latent states can be identified through Training across multiple environments. Thus, two different questions are answered.


Minor Issues:

Page 4, Line 196: Missing "E" in section title.

Page 5, Line 220: The symbol $\mathcal{X}$ is used before it is defined.

Page 5, Line 247: Missing word (probably "such").

Page 6, Line 299: Missing space.

Page 7, Line 371: Missing space.

Page 16, Line 884: "datastes"; also the sentence "A general overview…" does not make sense to me.

Page 16, Line 925: Missing space.

Page 16, Line 934: Missing space and missing dot.

It is not clear to me what the lower part of Figure 4 should tell as one can just see some samples of the datasets.
The term "mixing function" both describes the map $g: z_t\to x_t$ and the composition of $g$ and the transition step.

---

> ### Author Rebuttal · Authors · 2026-03-31
>
> Thank you for your thoughtful review. We first clarify a few factual points:
>
> - **"State discovery" vs. "representation alignment":** The inputs to our model are image sequences xt's (as illustrated in Figure 4, , where the lower part shows examples of the raw image observations fed to the model), not the physical state zt (never accessible during training). The two are related by the mixing function xt=g(zt). Crucially, perfectly fitting p(xt) does not guarantee recovery of the underlying latent states zt — this is precisely the identifiability problem we address. Our theoretical results (Theorems 1 and 2) establish that, when the model fits observations well, the learned latent variables are identifiable up to permutation, scaling, and translation of the true generative states. The experiments therefore validate state discovery, not merely representation alignment.
>
> - **Experimental validation of theoretical results and confusion for Fig 2 & 5:** We address identifiability at three levels: the latent environment variable c, the latent dynamics state zt, and the dynamics function. Our experiments directly validate theoretical claims for both c and zt. In Fig 2 and 5, identifiability results are displayed for both zt (blue) and c (red), as labeled in plot titles. Each plot compares ground-truth values (x-axis) and estimated values (y-axis): a commonly-adopted visualization to verify the relation between recovered and true latent parameters. Axis scales were omitted as the focus is on identifiable structure rather than absolute values, which we will add in revision. Under our framework, all zt satisfy the same identifiable relationship with ground-truth regardless of originating environment — hence plotted zt points are not differentiated by environment.
>
>     Additionally, we reported quantitative MCC metrics for both c and zt whenever ground truths are available (Table 1, 4, and 5), a metric widely adopted in identifiability literature (e.g., Khemakhem et al., 2020; Yao et al., 2022). The only exception was the real-world CMU datasets, where truths for c and zt were unavailable. The identifiability of latent dynamics function F is indirectly evidenced by prediction performance. Therefore, our theory and experiments address the same question.
>
> - **The choice of varying physical parameters:** We focused on physics parameters that directly affect the latent dynamics functions. Any parameters modulating the dynamics function without affecting the emission function can serve as the environment variable — to demonstrate this, we additionally provide results with varying mass ratio between the two pendulums (see Figure 3 in this  [anonymous link](https://anonymous.4open.science/api/repo/ICML26-B224/file/ICML_Rebuttal.pdf?v=6a0a4861)). Pendulum length L affects the emission function, so varying it would conflict with the assumption xt = g(zt) — adopted in most existing theoretical frameworks for identifiable latent dynamics. Allowing L to vary would necessitate modifying the theoretical framework, which we leave for future work.
>
> We now address the remaining questions:
>
> - **Definitions of the "mixing function":** Our model involves two distinct latent variables — zt and c, corresponding to two mixing functions: zt→xt and (z0, c)→x0:T. We acknowledge the paper does not clearly distinguish between the two uses and will clarify in revision.
>
> - **Conditions of Theorem 1:** Assumption 1 (noise distribution) is satisfied by standard Gaussian noise. Assumption 2 (mixing function injectivity) holds by construction in synthetic datasets. Assumption 3 (sufficient statistics) is trivially satisfied with our Gaussian prior where $T_{ij} = (c, c^2)$. Assumption 4 requires sufficient environment variability — the theoretical minimum is $2d+1$ environments, though this assumes idealized conditions. In practice, more environments are needed for reliable identifiability, as evidenced in Figure 2 of the [anonymous link](https://anonymous.4open.science/api/repo/ICML26-B224/file/ICML_Rebuttal.pdf?v=6a0a4861) with detailed results in our response to Reviewer Nqhj. Dimensions of $\mathbf{c}$ with negligible variation across environments cannot meaningfully be treated as environment variables, making their non-identifiability expected and natural — this also provides a useful practical criterion for assessing whether identifiability can be expected given an observation dataset.
>
> - **Finite-sample identifiability:** Theoretical identifiability under finite data remains an open problem — the vast majority of existing works rely on infinite data assumptions. To our knowledge, only one work [1] has analyzed finite-sample identifiability, specifically for contrastive learning. Finite-sample theory for VAE-based identifiable models remains an open problem for the community.
>
>     [1] Lyu, Qi, and Xiao Fu. "On finite-sample identifiability of contrastive learning-based nonlinear independent component analysis." ICML, 2022.

---

> > ### Author Rebuttal · Reviewer_1Jb2 · 2026-04-03
> >
> > I thank the authors for their detailed comments on my concerns and their efforts for providing additional results. They have helped me understand points better, which is why I believe my current score accurately reflects the papers contribution.

---

### Official Review · Reviewer_UStq · 2026-03-13

**Soundness:** 3
**Presentation:** 3
**Significance:** 3
**Originality:** 3
**Overall Recommendation:** 4
**Confidence:** 4

**Summary:**

A  latent variable recovery method with identification theory and empirical gains.

**Compliance With Llm Reviewing Policy:**

Affirmed.

**Final Justification:**

I have read authors' rebuttal. Although my concern about clustering-based approach is not addressed, I think this still provides some insights where we may use some context examples as auxiliary information. So I maintain my score.

**Key Questions For Authors:**

Please see above.

**Limitations:**

yes

**Strengths And Weaknesses:**

Pros:

1. I agree with authors that discrete environmental labels may not always exist and in some cases, some examples from the environment should be accessible and utilized to infer the label. This is novel.

2. It provides a method to infer the environmental label and shows the identifiability of the label and corresponding latent variables.

3. It also provides a practical implementation of the method and quantitative results are provided to demonstrate the effectiveness.

Cons:

1. How many context samples are needed to infer the label in practice? What is difference compared to clustering and then apply TDRL method? (Some methods also show identifiabity of discrete latent variables without label, can we combine them?)

---

> ### Author Rebuttal · Authors · 2026-03-31
>
> Thank you for your positive feedback and constructive questions.
>
> - **Number of context samples $k$:** In all reported experiments, $k$ was set to 9. We provide additional results on the Pendulum dataset with $k$ varying from 0 to 9 (Figure 1, [anonymous link](https://anonymous.4open.science/api/repo/ICML26-B224/file/ICML_Rebuttal.pdf?v=6a0a4861)), where $k=0$ denotes the setting where $\mathbf{c}$ is inferred from the query sample itself (as a reference). As $k$ decreases, identifiability of $\mathbf{c}$ degrades gradually, yet even at $k=1$ a meaningful identifiable structure is retained, suggesting that a single context sample already provides useful information about the environment.
>
> | | $k=0$ | $k=1$ | $k=3$ | $k=5$ | $k=7$ | $k=9$ |
> |---|---|---|---|---|---|---|
> | MCC($z$) | 0.9916 | 0.9950 | 0.9986 | 0.9982 | 0.9975 | 0.9985 |
> | MCC($c$) | 0.8918 | 0.9331 | 0.9427 | 0.9528 | 0.9734 | 0.9779 |
>
> - **Comparison with clustering-based approaches:** Existing clustering-based methods such as VaDE [1] and MFCVAE [2] learn a discrete label $u$ through clustering, and recent theoretical work [3] has shown that such deep clustering models can also achieve identifiability. Replacing the manually provided auxiliary variable $u$ required by related works (TDRL and IDF) with a clustering-derived counterpart is an interesting direction worth exploring. However, existing deep clustering methods have primarily been demonstrated on datasets with well-defined cluster structures (e.g., MNIST), where clustering targets correspond to digit identity or writing style. Whether such approaches transfer effectively to physical dynamical systems, where environment variables are continuous and reflect underlying physical parameters, remains an open question with no existing work to our knowledge. The performance of clustering-derived labels in OOD settings is also unclear. We agree this is a promising direction for future investigation.
>
> [1] Jiang et al., "Variational deep embedding." IJCAI, 2017.
> [2] Falck et al., "Multi-facet clustering variational autoencoders." NeurIPS, 2021.
> [3] Kivva et al., "Identifiability of deep generative models without auxiliary information." NeurIPS, 2022.

---

> > ### Author Rebuttal · Reviewer_UStq · 2026-04-02
> >
> > I have read authors' rebuttal. Although my concern about clustering-based approach is not addressed, I think this still provides some insights where we may use some context examples as auxiliary information. So I maintain my score.

---

### Official Review · Reviewer_Nqhj · 2026-03-13

**Soundness:** 3
**Presentation:** 3
**Significance:** 3
**Originality:** 3
**Overall Recommendation:** 5
**Confidence:** 2

**Summary:**

Meta-iLaD addresses identifiability in locally-stationary latent dynamics $F(z_{<t}; c)$ where c is a dynamics environment variable. Prior identifiable methods (TDRL, IDF) require predefined environment labels u, limiting OOD generalization. Meta-iLaD replaces labels with few-shot context samples via a feedforward meta-learner, and establishes simultaneous identifiability for the latent state $z_t$, environment variable $c$, and dynamics function $F$. The model uses a Neural ODE for dynamics, a factorized Gaussian conditional prior for $c$ inferred from context samples, and is trained via a meta-learning ELBO with TC regularization and warm-up strategies. Experiments on three synthetic physics systems (with ground truth) and CMU MoCAP demonstrate strong reconstruction, prediction, and OOD performance.

**Compliance With Llm Reviewing Policy:**

Affirmed.

**Key Questions For Authors:**

- What value of k is used in each experiment, and how does performance degrade as k → 1?
- See weaknesses

**Limitations:**

yes

**Strengths And Weaknesses:**

## Strength
1. The reliance on predefined environment labels in prior identifiable dynamics work (TDRL, IDF) is a real practical bottleneck. Replacing labels with few-shot context samples enables OOD generalization that label-dependent methods structurally cannot achieve. The three-level identifiability hierarchy (z_t, F, c) is a useful organizing framework.
2. The Ablation study in Table 2 directly tests the key claimed advantage and establishes that context-sample performance (MCC(c) ≈ 0.98) nearly matches the oracle true-parameter condition (MCC(c) ≈ 0.99). Table 4 cleanly disentangles TC regularization (affects MCC(c)) from warm-up (affects MCC(z_t)).
3. The simultaneous reporting of MSE, MCC(z_t), and MCC(c) on synthetic systems with known ground truth is methodologically sound. The key empirical finding — that MCC(c) differences between methods are inconsequential during reconstruction but translate to large MSE differences during prediction — effectively supports the motivation that identifiability of c and F matters for forecasting.



## Weakness
1. The number of context samples k is not clearly reported in the main text, nor is there sensitivity analysis over k. This is a critical hyperparameter: too few samples may fail to identify c; too many may be impractical.
2. All synthetic systems have c ∈ R^1 or c ∈ R^2. This is the minimal non-trivial case. No evidence is provided that identifiability holds when latent dimensionality increases (d=5, d=10). Theorem 1's requirement of 2d+1 environments is never stress-tested.

---

> ### Author Rebuttal · Authors · 2026-03-31
>
> Thank you for your thorough and constructive review. We address each weakness below.
>
> - **Number of context samples $k$:** The value of $k$ was set to 9 in all reported experiments. We provide sensitivity results on the Pendulum dataset as $k$ decreases from 9 to 1 in the table below and in Figure 1 of the [anonymous link](https://anonymous.4open.science/api/repo/ICML26-B224/file/ICML_Rebuttal.pdf?v=6a0a4861). Here $k=0$ denotes the baseline setting where $\mathbf{c}$ is inferred from the query sample itself. As $k$ decreases, identifiability of $\mathbf{c}$ degrades gradually, yet even at $k=1$ a meaningful identifiable structure is retained, suggesting that a single context sample already provides useful information about the environment.
>
> | | $k=0$ | $k=1$ | $k=3$ | $k=5$ | $k=7$ | $k=9$ |
> |---|---|---|---|---|---|---|
> | MCC($z$) | 0.9916 | 0.9950 | 0.9986 | 0.9982 | 0.9975 | 0.9985 |
> | MCC($c$) | 0.8918 | 0.9331 | 0.9427 | 0.9528 | 0.9734 | 0.9779 |
>
> - **Stress-testing the $2d+1$ environment requirement:** We provide results on the Pendulum dataset as the number of training environments increases from 5 to 100, reported below and in Figure 2 of the [anonymous link](https://anonymous.4open.science/api/repo/ICML26-B224/file/ICML_Rebuttal.pdf?v=6a0a4861). As expected, MCC($\mathbf{c}$) improves progressively as the number of environments increases. While Theorem 1 establishes $2d+1$ as the theoretical minimum, this bound assumes idealized conditions (e.g., exact distribution matching, infinite data) that are difficult to satisfy in practice. In practice, a substantially larger number of environments is needed to achieve reliable identifiability of $\mathbf{c}$, as reflected in our results.
>
> | Env num ($d=2$) | 5 | 10 | 50 | 100 |
> |---|---|---|---|---|
> | MCC($z$) | 0.9993 | 0.9976 | 0.9972 | 0.9989 |
> | MCC($c$) | 0.7446 | 0.9317 | 0.9428 | 0.9782 |
>
> - **Identifiability with higher-dimensional $\mathbf{c}$:** Identifying higher-dimensional environment variables is challenging because, as the dimension of $\mathbf{c}$ increases, it becomes increasingly difficult for its mapping to the observations $x_{0:T}$ to remain injective (condition 2 of Theorem 1). More specifically, $\mathbf{c}$ contributes to observations $x_{0:T}$ through two complex steps: $\mathbf{c}$ shapes the entire latent trajectory $z_{0:T}$ through the dynamics function $\mathcal{F}$, which is then observed through the emission function $g$. The influence of individual dimensions of $\mathbf{c}$ on the final observations can therefore be weak or entangled with other dimensions, violating the assumption of a bijective mapping between $\mathbf{c}$ and observations $x_{0:T}$. When this occurs, identifiability of those dimensions becomes inherently difficult — a fundamental challenge shared by any method for multi-parameter dynamical systems, not specific to ours. In preliminary experiments with multi-parameter systems, we observe that some dimensions of $\mathbf{c}$ are well-recovered while others are not, consistent with this analysis. We consider the theoretical and empirical investigation of high-dimensional $\mathbf{c}$ an important direction for future work.

---

> > ### Author Rebuttal · Reviewer_Nqhj · 2026-04-05
> >
> > Thank you for your thorough response. Having carefully re-considered the experimental and theoretical results in the paper and the rebuttal, I stand by my original score.

---

### Decision · Program_Chairs · 2026-04-30

**Decision:**

Accept (regular)

**Comment:**

This paper proposes a meta-learning framework for identifiable latent dynamics that eliminates the reliance on predefined environment labels. The proposed method replaces label conditioning with a feedforward meta-learner that infers environment variables from few-shot context samples. The effectiveness of the proposed method is demonstrated in the experiments.
The reviewers broadly agree on the novelty and soundness of the contribution. The rebuttal adequately addressed the major concerns, including sensitivity to the number of context samples, stress-testing,  and the distinction between state discovery and representation alignment. Some remaining issues (e.g., computational cost, sensitivity to assumption) are acknowledged limitations rather than fundamental flaws. The presentation can be improved. A comparison with clustering-based approaches or its discussion would strengthen the paper.